# BALF: BUDGETED ACTIVATION-AWARE LOW-RANK FACTORIZATION FOR FINE-TUNING-FREE MODEL COMPRESSION

## ABSTRACT

Neural network compression techniques typically require expensive fine-tuning or search procedures, rendering them impractical on commodity hardware. Inspired by recent LLM compression research, we present a general activation-aware factorization framework that can be applied to a broad range of layers. Moreover, we introduce a scalable budgeted rank allocator that allows flexible control over compression targets (e.g., retaining 50% of parameters) with no overhead. Together, these components form BALF, an efficient pipeline for compressing models without fine-tuning. We demonstrate its effectiveness across multiple scales and architectures, from ResNet-20 on CIFAR-10 to ResNeXt-101 and vision transformers on ImageNet, and show that it achieves excellent results in the fine-tuning-free regime. For instance, BALF reduces FLOPs on ResNeXt-101 by 45% with only a 1-percentage-point top-1 accuracy drop.

## 1 INTRODUCTION

Deep learning models achieve cutting-edge results in various domains (LeCun et al., 2015), yet their computational and memory demands limit deployment feasibility, motivating the need for model compression methods. Among these, factorization is a strong candidate. Traditional techniques use singular value decomposition (SVD)-based methods to minimize the discrepancy between original and compressed parameters (Jaderberg et al., 2014; Hua et al., 2023). Nonetheless, such techniques often require fine-tuning and/or costly search procedures. For large networks, this is impractical on standard hardware.

Recently, the rise in popularity of large language models (LLMs) has led to the exploration of activation-aware factorization techniques, which seek to lessen the (expected) output distortion at each layer (Wang et al., 2025c;b; Chen et al., 2021), often reducing the need for fine-tuning. These techniques predominantly focus on fully connected layers, which are central to attention-based LLMs (Vaswani et al., 2017).

Here, we propose a unified framework that builds on Wang et al. (2025c) and is compatible with a broad class of layers. Moreover, we introduce a scalable rank allocator that determines how much to compress each layer based on a Lagrangian relaxation. Together, these form BALF, a complete and general fine-tuning-free compression pipeline. Our main contributions are:

- **Unified and principled framework.** We extend activation-aware low-rank factorization (Wang et al., 2025c;b) to a broad class of layers, including (but not limited to) (possibly grouped) convolutional layers. Moreover, we rework the activation-aware theory from scratch to handle activations that may lie in subspaces (i.e., linearly redundant).
- **Budget-aware allocation.** We introduce a zero-overhead rank allocator that meets user-specified FLOPs or parameter count budgets.
- **Practicality.** BALF runs end-to-end efficiently on commodity hardware (e.g., on an RTX 2070 laptop, it takes less than four minutes to compress most models we tested), and avoids expensive hyperparameter sweeps.
- **Extensive evaluation.** Across vision models from ResNet-20 (He et al., 2016) on CIFAR-10 (Krizhevsky, 2009) to ResNeXt-101 (Xie et al., 2017) and ViT-B/16 (Dosovitskiy et al.,

2021) on ImageNet (Deng et al., 2009), our method achieves strong accuracy-compression trade-offs in the fine-tuning-free regime.

## 2 RELATED WORK

Deep learning model compression has been extensively studied. The principal categories of compression techniques include quantization, which employs low-bit storage and operations (Jacob et al., 2018; Frantar et al., 2023; Gholami et al., 2021); pruning, which involves the elimination of parameters or parameter groups from the model (Hoefler et al., 2021; Frantar & Alistarh, 2022); and factorization, which divides model layers into smaller sub-layers to lower overall memory usage and computational demands (Jaderberg et al., 2014; Ben Noach & Goldberg, 2020). This paper focuses on the factorization approach.

Various factorization methods have been extensively studied over time. Among these, the SVD (Golub & Van Loan, 2013) stands out as a straightforward approach. It is directly applicable to fully connected layers, such as those found in transformer architectures, consisting mainly of matrix multiplications.

In convolutional neural networks (CNNs), factorization is usually performed by reshaping parameter tensors into matrices and subsequently decomposing them (Yang et al., 2020), with an alternative research direction employing other kinds of decompositions (Denton et al., 2014; Yin et al., 2021; Phan et al., 2020; Liebenwein et al., 2021; Lebedev et al., 2014).

Existing factorization approaches often require fine-tuning and/or compute-intensive search (Idelbayev & Carreira-Perpiñán, 2020; Yu & Bouganis, 2023). The impossibility of running these on common hardware motivates the need for lighter methods. For example, pruning methods that explicitly aim to be efficient in this regard include Narshana et al. (2023); Murti et al. (2023); Chen et al. (2024); Zhang et al. (2024); Wang et al. (2025a), whereas fine-tuning-free factorization methods are less common—for instance, Yu & Bouganis (2023). Liebenwein et al. (2021); Idelbayev & Carreira-Perpiñán (2020) also report some results in the fine-tuning-free regime.

Recent factorization research, motivated by the computational demands of LLMs, leverages intermediate model features to identify additional redundancies suitable for decomposition (Chen et al., 2021; Wang et al., 2025c;b; Qinsi et al., 2025; Yuan et al., 2025). These methods use a calibration dataset to characterize the distribution of feature tensors and then use that information to guide parameter decompositions. However, they primarily focus on decomposing fully connected layers in the language domain, often overlooking other settings.

In contrast, BALF is able to perform activation-aware compression on a general class of layers in a principled way, including convolutional layers. Together with our efficient rank allocation strategy, we obtain excellent performance in the no-fine-tuning regime in minutes without requiring expensive hardware.

## 3 PRELIMINARIES AND NOTATION

**The singular value decomposition.** Given a matrix $\boldsymbol{A} \in \mathbb{R}^{M \times N}$, we denote its SVD by $\boldsymbol{A} = \boldsymbol{U}\boldsymbol{\Sigma}\boldsymbol{V}^T$, where $\boldsymbol{U} \in \mathbb{R}^{M \times M}$ and $\boldsymbol{V} \in \mathbb{R}^{N \times N}$ are orthogonal matrices, and $\boldsymbol{\Sigma} = \mathrm{diag}(\boldsymbol{\sigma}; (M, N))$ is a diagonal matrix with its singular values. Another concept of interest is the $P$-truncated SVD, which is defined by taking only the $P$ largest singular values and their associated singular vectors. In particular, we define $\mathcal{T}_P^{\mathrm{SVD}}(\boldsymbol{A}) = \boldsymbol{U}_{:,:P}\boldsymbol{\Sigma}_{:P,:P}(\boldsymbol{V}_{:,:P})^T$. It can be thought of as an operator that projects a matrix to its rank-$P$ approximation via SVD.

**Uncentered whitening.** Data whitening (Kessy et al., 2018) (also termed data sphering) is a widely used technique in machine learning and statistics. Here, we are interested in a special case called uncentered whitening. This process transforms data so that its second-moment matrix becomes the identity matrix. Formally, we define an uncentered whitening matrix (UWM) $\boldsymbol{M} \in \mathbb{R}^{I \times I}$ for the data $\boldsymbol{X} \in \mathbb{R}^{N \times I}$ as a matrix such that $\boldsymbol{M}^T \boldsymbol{X}^T \boldsymbol{X} \boldsymbol{M} = N \overline{\boldsymbol{I}}_R$, where $\overline{\boldsymbol{I}}_R \in \mathbb{R}^{I \times I}$ is the identity on the first $R$ coordinates and zero elsewhere, and $R$ is the rank of $\boldsymbol{X}$. We further restrict the definition by requiring that $\mathrm{row}(\boldsymbol{X}) = \mathrm{range}(\boldsymbol{M})$, which in turn forces the last $I - R$ columns

of $M$ to be zero and makes $X = XMM^+$; this assumption and these consequences will be used in later results.

**Other notation and conventions.** We refer to general tensors as $\mathbf{X}$. When some tensor is known to be a matrix or a vector, we may use $X$ and $x$, respectively. For matrices, we denote the Moore-Penrose inverse (also termed the pseudoinverse) of $M$ by $M^+$, and the square root of the pseudoinverse by $M^{+1/2}$.

Deep learning frameworks, such as PyTorch (Ansel et al., 2024), usually include "reshape" and "permute" operators, denoted in this work by $\mathsf{reshape}\,(\mathbf{X}; (s_1, \ldots, s_n))$ and $\mathsf{permute}\,(\mathbf{X}; (\pi_1, \ldots, \pi_n))$, respectively.

When talking about network layers, for simplicity of notation, unless specified, we drop the layer identifiers and denote the computation carried out by a layer by $f(\mathbf{X}; \mathbf{W})$, where $\mathbf{X}$ denotes the layer's input and $\mathbf{W}$ the parameters specific to that layer. In the case of the first layer, $\mathbf{X}$ denotes the input to the network (e.g., a multi-channel 2D signal). In general, when discussing factorization techniques, we ignore bias terms because they are unaffected by factorization; only the primary operations (e.g., convolutions or matrix multiplications) are modified.

Throughout this work, $\|\cdot\|_F$ refers to the tensor Frobenius norm, i.e., $\|\cdot\|_F = \|\mathsf{flatten}(\cdot)\|_2$.

We denote the identity function by $\mathrm{id}$. Moreover, when discussing convolutional layers, $B$ will denote the batch size; $H_x, W_x, C_x$ the height, width, and channels of the input (when $x = i$) and output (when $x = o$); and $H_k, W_k$ the spatial size of the kernel. In this context, we also refer to $G$ as the number of groups in the convolutional layer (Xie et al., 2017).

## 4 OUR FRAMEWORK

Before diving into the factorization scheme, we first restrict our discussion to a particular class of layers that can be expressed as a combination of matrix multiplications and auxiliary functions.

**Definition 1.** *A layer $f$ is $(\overline{O}, \overline{I}, \overline{P})$-expressible if it can be expressed as $f(\mathbf{X}; \mathbf{W}) = \overline{O}(\overline{I}(\mathbf{X})\overline{P}(\mathbf{W}))$, where $\overline{O}$ and $\overline{P}$ are compositions of reshape and permute operators, and $\overline{I}$ is a linear operator. $\overline{O}$ takes a batch of matrices in the form of a third-order tensor and outputs a tensor (possibly a matrix), whereas $\overline{P}$ and $\overline{I}$ both take a tensor (possibly a matrix) and output a batch of matrices in the form of a third-order tensor of compatible shapes. In particular, $\overline{O} : \mathbb{R}^{G \times N \times O} \mapsto \cdot$, $\overline{I} : \cdot \mapsto \mathbb{R}^{G \times N \times I}$, and $\overline{P} : \cdot \mapsto \mathbb{R}^{G \times I \times O}$.*

$G$ stands for the number of groups. In the special case where $G = 1$, we may simply refer to matrices instead of batches containing a single matrix. Note that in the preceding definition, matrix multiplication is batched, i.e., $G$ independent matrix multiplications are carried out. The necessity of batch semantics in the group dimension will become apparent later. This condition is satisfied by popular layers. In what follows, layers are assumed to operate in batched mode, with $B$ denoting the batch size. We refer to $N$ as the outer dimension.

**Example 1.** *Fully connected layers are $(\mathrm{id}, \mathrm{id}, \mathrm{id})$-expressible.*

In some architectures (e.g., transformers), linear layers receive inputs with multiple leading independent dimensions—for example, a tensor of shape $(B, L, D)$. In those cases, we flatten the leading dimensions, reshaping the input to $(BL, D)$ and restoring the original shape afterward. We now turn our attention to convolutional layers. Although an ungrouped convolution is a special case of a (possibly grouped) convolution, we present it separately for clarity.

**Example 2.** *Ungrouped convolutional layers are $(\overline{O}, \overline{I}, \overline{P})$-expressible, with*

$$\overline{O}(\mathbf{Y}) = \mathsf{permute}(\mathsf{reshape}(\mathbf{Y}; (B, H_o, W_o, C_o)); (0, 3, 1, 2)),$$

$$\overline{I}(\mathbf{X}) = \mathsf{reshape}(\mathsf{im2col}(\mathbf{X}); (BH_oW_o, C_iH_kW_k)),$$

$$\overline{P}(\mathbf{W}) = \mathsf{reshape}(\mathbf{W}; (C_o, C_iH_kW_k))^T,$$

where $\mathsf{im2col} : \mathbb{R}^{B \times C_i \times H_i \times W_i} \to \mathbb{R}^{B \times H_oW_o \times C_iH_kW_k}$ maps a batch of 2D signals with $C_i$ channels into a matrix (see Chellapilla et al. (2006)). The PyTorch equivalent is called unfold[1]. It encapsulates

---

[1] https://docs.pytorch.org/docs/stable/generated/torch.nn.Unfold.html

the convolution hyperparameters—kernel size, stride, padding, and dilation—but not the number of groups. For brevity, we omit the encapsulated hyperparameters from the notation.

On the other hand, a grouped convolution can be viewed as $G$ separate, parallel convolutions. Each one takes a disjoint slice of $C_i/G$ input channels and produces $C_o/G$ output channels. Concatenating the $G$ outputs recovers the full $C_o$ channels (Krizhevsky et al., 2012; Xie et al., 2017). This also fits our framework.

**Example 3.** *Grouped convolutional layers are $(\overline{O}, \overline{I}, \overline{P})$-expressible, with*

$$\overline{O}(\mathbf{Y}) = \text{permute}(\text{reshape}(\text{permute}(\mathbf{Y}; (0, 2, 1)); (C_o, B, H_o, W_o)); (1, 0, 2, 3)),$$

$$\overline{I}(\mathbf{X}) = \text{permute}(\text{reshape}(\text{im2col}(\mathbf{X}); (BH_oW_o, G, (C_i/G)H_kW_k)); (1, 0, 2)),$$

$$\overline{P}(\mathbf{W}) = \text{permute}(\text{reshape}(\mathbf{W}; (G, C_o/G, (C_i/G)H_kW_k)); (0, 2, 1)).$$

Additionally, Example 2 and Example 3 have trivial extensions to higher-dimensional convolutions, achieved by replacing im2col with its higher-dimensional counterparts.

Low-rank factorization applies naturally to linear layers by decomposing their weight matrices. The main idea of our framework is to extend the notion of factorization to $(\overline{O}, \overline{I}, \overline{P})$-expressible layers by exploiting their equivalence to (possibly batched) matrix multiplications.

Let us focus on linear layers for a moment. Recall that a low-rank matrix (or its low-rank projection) can be expressed as a combination of a tall and a wide matrix. $\mathcal{T}_P(\mathbf{W})$ is defined to be a generic operator that returns a low-rank projection of $\mathbf{W}$. If we fix $P$ and approximate $\mathbf{X}\mathbf{W} \approx \mathbf{X}\mathcal{T}_P(\mathbf{W}) = (\mathbf{X}\mathbf{W}_0)\mathbf{W}_1$, with $\mathbf{W} \in \mathbb{R}^{I \times O}, \mathbf{W}_0 \in \mathbb{R}^{I \times P}, \mathbf{W}_1 \in \mathbb{R}^{P \times O}$, for sufficiently small $P$, the storage and compute requirements will be less than those of the original layer. By using their $(\overline{O}, \overline{I}, \overline{P})$ representation, we can extend this to, for instance, convolutional layers. It can be verified that ($\overline{P}^{-1}$ denotes the inverse) $\mathbf{X} * \mathbf{W} \approx \mathbf{X} * \overline{P}^{-1}\left(\mathcal{T}_P(\overline{P}(\mathbf{W}))\right) = (\mathbf{X} * \mathbf{W}_0) * \mathbf{W}_1$ for some $\mathbf{W}_0 \in \mathbb{R}^{P \times C_i \times H_k \times W_k}, \mathbf{W}_1 \in \mathbb{R}^{C_o \times P \times 1 \times 1}$, obtained from decomposing $\overline{P}(\mathbf{W}) \in \mathbb{R}^{C_o \times C_i H_k W_k}$. Again, with sufficiently small $P$, this results in storage and compute savings.

**SVD-based factorization.** SVD-based factorization is just a specific way of obtaining two low-rank factors from a matrix. In particular, as per the Eckart–Young–Mirsky (EYM) theorem (Golub & Van Loan, 2013), it is the optimal factorization scheme in terms of parameter distortion, defined as

$$\ell^{\text{param}}(\mathbf{W}, P) = \|\mathbf{W} - \mathcal{T}_P^{\text{SVD}}(\mathbf{W})\|_F^2. \tag{1}$$

In order to simplify the notation, we define $\mathcal{T}_P^{\text{SVD}}(\mathbf{W}) \equiv \overline{P}^{-1}\left(\mathcal{T}_P^{\text{SVD}}(\overline{P}(\mathbf{W}))\right)$. In other words, we extend the matrix low-rank projection notion to arbitrary tensors that represent the weights of $(\overline{O}, \overline{I}, \overline{P})$-expressible layers. In the case where $G > 1$, we decompose each group $\overline{P}(\mathbf{W})_g$ separately, with the constraint that the per-group rank is the same across groups. This latter constraint has practical reasons, which are discussed in Appendix A.3.1.

**General low-rank projections.** Note that there are many ways one can obtain low-rank factors from a matrix (or a tensor, with our defined extension). A general class of operators can be defined (for a group index $g$) as

$$\mathcal{T}_P^D\left(\overline{P}(\mathbf{W})_g\right) = \arg \min_{\text{rank}(\overline{P}(\widehat{\mathbf{W}})_g) \leq P} D(\overline{P}(\mathbf{W})_g, \overline{P}(\widehat{\mathbf{W}})_g), \tag{2}$$

where $D$ is some penalty function, and $\widehat{\mathbf{W}}$ is the approximate tensor. Traditional SVD-based decomposition corresponds to the case where $D$ is the Frobenius norm of the (reshaped) parameters, i.e., it penalizes distance in parameter space (see Equation (1)). We use $\mathcal{T}_P^D(\mathbf{W})$ to also denote batched computation, where each group's matrix is decomposed individually. As before, we omit $\overline{P}$ to simplify notation.

### 4.1 ACTIVATION-AWARE LOW-RANK PROJECTIONS

**Motivation.** A more natural proxy for task performance is the (expected) distortion in each layer's outputs. In general, though, we have a calibration dataset (which can be the entire training dataset,

a subset of it, etc., and is assumed to consist of i.i.d. samples), used to gather layer activations $(\mathbf{X}^{(i)})_{i=1}^{B}$, which we then use to estimate the preceding expectation. The (empirical) average activation distortion for a layer $f$ is defined as

$$\ell^{\text{activ}}(\mathbf{W}, P) = \frac{1}{B} \sum_{i=1}^{B} \|f(\mathbf{X}^{(i)}; \mathbf{W}) - f(\mathbf{X}^{(i)}; \mathcal{T}_P(\mathbf{W}))\|_F^2 = \frac{1}{B} \|f(\mathbf{X}; \mathbf{W}) - f(\mathbf{X}; \mathcal{T}_P(\mathbf{W}))\|_F^2. \quad (3)$$

Note that the last equality holds when forming a "batch" with the list $(\mathbf{X}^{(i)})_{i=1}^{B}$. In what follows, our analysis is performed in terms of the intermediate activations $\mathbf{X}$ obtained with the calibration dataset, with the expectation that the results carry over to the test dataset. We now aim to find a low-rank projection operator that minimizes Equation (3).

**Which projection is best?** In order to answer that question, we first need to define what "best" should mean. Without loss of generality, fix a group $g$. Clearly, $\text{rank}\left(\overline{I}(\mathbf{X})\mathcal{T}_P(\mathbf{W})\right) \leq P$. Moreover, by the EYM theorem, the best rank-$P$ approximation of the output of $f$ in terms of the Frobenius norm is $\mathcal{T}_P^{\text{SVD}}\left(\overline{I}(\mathbf{X})\overline{P}(\mathbf{W})\right)$. Hence, a reasonable notion of an ideal scheme is one that factorizes the parameters in a way that is equivalent to factorizing the outputs of the layer; something better is impossible.

**Definition 2.** *We call a scheme $\mathcal{T}_P(\cdot)$ optimal if its distortion is equivalent to directly low-rank truncating the outputs of a layer, i.e., an optimal scheme satisfies*

$$\|\overline{I}(\mathbf{X})\overline{P}(\mathbf{W}) - \mathcal{T}_P^{\text{SVD}}\left(\overline{I}(\mathbf{X})\overline{P}(\mathbf{W})\right)\|_F = \|\overline{I}(\mathbf{X})\overline{P}(\mathbf{W}) - \overline{I}(\mathbf{X})\mathcal{T}_P\left(\overline{P}(\mathbf{W})\right)\|_F.$$

Recall that each group is factorized separately. Note that a scheme is optimal if and only if it satisfies Equation (2) with the distance function set to the output distortion. In particular, SVD-LLM (Wang et al., 2025c) introduces a specific way of "talking in terms of activation distortion" for linear layers, and it is optimal in the sense of Definition 2. A more general version of their low-rank activation-aware projection method, from the lens of our framework, can be defined as follows. For any $(\overline{O}, \overline{I}, \overline{P})$-expressible function $f$, let

$$\mathcal{T}_P^{\text{AA}}(\mathbf{W}) = \overline{P}^{-1}\left(\mathbf{M}\mathcal{T}_P^{\text{SVD}}\left(\mathbf{M}^+\overline{P}(\mathbf{W})\right)\right),$$

where $\mathbf{M}$ is a batch of whitening matrices for $\overline{I}(\mathbf{X})$ (where each batch pertains to a group, and the matrix multiplication is batched over the groups dimension), and the pseudoinverse is performed over each group.

For the layers and their corresponding $(\overline{O}, \overline{I}, \overline{P})$-tuples mentioned earlier, this corresponds, as discussed in previous examples, to two sequential sub-layers (e.g., two linear or convolutional layers) that yield computational gains with sufficiently low $P$. More details are given in Appendix A. Additionally, our scheme remains optimal, as claimed in Theorem 1.

**Theorem 1.** $\mathcal{T}_P^{\text{AA}}(\cdot)$ *is optimal in the sense of Definition 2.*

Moreover, the activation distortion that follows from using our framework is readily computable from the artifacts obtained during compression, as shown in Theorem 2.

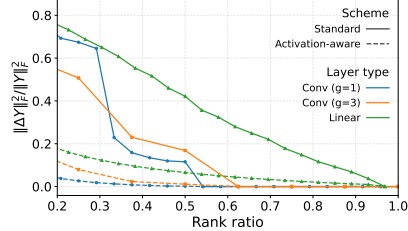

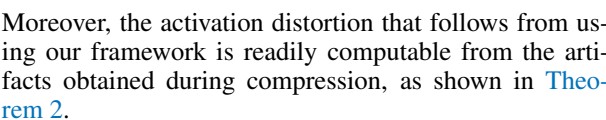

Figure 1: Normalized output distortion (in terms of squared Frobenius norm) with $\mathcal{T}_P^{\text{AA}}(\cdot)$ and $\mathcal{T}_P^{\text{SVD}}(\cdot)$ for different values of $P$, across different layer types.

**Theorem 2.** *Assume that $f(\mathbf{X}; \mathbf{W})$ is $(\overline{O}, \overline{I}, \overline{P})$-expressible. Then, the activation distortion (as defined in Equation (3)) incurred when low-rank projecting the parameters with $\mathcal{T}_P^{\text{AA}}(\cdot)$ is $\frac{N}{B} \sum_{g=1}^{G} \sum_{i=P+1}^{U} \sigma_{g,i}^2$, where $\boldsymbol{\sigma}$ denotes the batched singular values of $\mathbf{M}^+\overline{P}(\mathbf{W})$.*

Recall that $B$ is the batch size (number of calibration samples), and $N$ is the outer dimension (see the discussion below Definition 1). The main result of Wang et al. (2025c) follows as a corollary.

**A simple experiment.** We compare the output distortion of our activation-aware projection with that of standard SVD. In particular, we select three layers of different types, truncate them to various rank ratios, and measure the resulting output distortions on 1024 CIFAR-10 test images (using 1024 training images for calibration). As shown in Figure 1, activation-aware factorization yields significantly lower distortion.

## 4.2 COMPUTING THE WHITENING MATRICES

We compute the whitening matrices using an eigendecomposition of the transformed features. Suppose we are given a group $g$. Let $\boldsymbol{X} = \overline{I}(\mathbf{X})_g \in \mathbb{R}^{N \times I}$. We can then compute the eigenvectors and eigenvalues of $\frac{1}{N} \boldsymbol{X}^T \boldsymbol{X}$: $\boldsymbol{V}$ and $\boldsymbol{\Lambda}$, respectively, and set

$$\boldsymbol{M} = \boldsymbol{V} \boldsymbol{\Lambda}^{+1/2}, \quad \boldsymbol{M}^+ = \boldsymbol{\Lambda}^{1/2} \boldsymbol{V}^T.$$

This procedure is applied (in vectorized form) independently for each group. Second moments can be accumulated incrementally over minibatches, so the method is memory-efficient. Additional implementation details are provided in Appendix B.

## 4.3 BUDGETED RANK ALLOCATION

Apart from offering output distortion guarantees, this framework is interpretable in the sense that we can measure the output distortion without additional steps (see Theorem 2). Can that be used to efficiently choose the rank to which each layer should be truncated?

**The rank allocation problem.** We now turn to the question of how to select the ranks to retain in each layer, $\{P_l\}_{l=1}^L$. The naive option is to choose a uniform percentage of rank per layer (Wang et al., 2025c). A more reasonable option, used in the past for the traditional SVD scheme, is the energy-based singular value pruning criterion (Yang et al., 2020; Liebenwein et al., 2021). While more expensive approaches have been explored, such as training-based rank learning (Idelbayev & Carreira-Perpiñán, 2020; Qinsi et al., 2025) or NAS-based search (Yu & Bouganis, 2023), our main goal is to find a scalable and efficient method. We can define the analog of the energy metric in our setting as follows (it reads as the energy retained in layer $l$ when keeping up to rank $P_l$):

$$E_l(P_l) = 1 - \frac{\|f_l(\mathbf{X}^{l-1}; \mathbf{W}^l) - f_l(\mathbf{X}^{l-1}; \mathcal{T}_{\hat{P}_l}^{\text{AA}}(\mathbf{W}^l))\|_F^2}{\|f_l(\mathbf{X}^{l-1}; \mathbf{W}^l)\|_F^2} = \frac{\sum_{g=1}^G \sum_{i=1}^{P_l} \boldsymbol{\sigma}_{g,i}^2}{\sum_{g=1}^G \sum_{i=1}^{U_l} \boldsymbol{\sigma}_{g,i}^2},$$

where $l$ denotes the layer index, $U_l$ the total number of singular values (per group), and $P_l$ the number retained (per group). Energy-based selection then consists of picking the lowest rank such that a certain energy threshold (manually defined by the user) is retained. We propose instead to solve an optimization problem that maximizes the total retained energy globally:

$$\max_{P_1,\ldots,P_L} \sum_{l=1}^L E_l(P_l), \quad \text{subject to } \sum_{l=1}^L C_l(P_l) \le C_{\max} \quad \text{and} \quad P_l \in \{1,\ldots,U_l\},$$

where $C_l(P_l)$ is some measure of complexity (e.g., FLOPs or number of parameters when keeping up to rank $P_l$ on layer $l$), $U_l$ is the total number of singular values for a group on layer $l$, and $C_{\max}$ is the model complexity budget. Our allocator supports FLOPs and the number of parameters as complexity targets. The optimization problem is an instance of the multiple-choice knapsack problem, and it can be formulated as a binary linear program. Unfortunately, it is known to be NP-hard (for a review, see Szkaliczki (2025)), which motivates the need for an alternative.

**A Lagrangian relaxation.** We solve this limitation by using a Lagrangian relaxation of the problem, whose time complexity is $O((I+D+1)\sum_l U_l)$, where $I$ stands for "number of iterations" and $D$ depends on the network. In practice, using 300 iterations yields excellent results while taking less than 0.2 seconds, even for the larger models with which we experimented. We provide additional details in Appendix C.1.

**Advantages.** Our selection approach has several advantages: (1) it has virtually zero additional overhead and does not query the model, unlike brute-force search methods, which can take hours (e.g., Yu & Bouganis (2023)), and (2) it provides explicit control over a complexity budget. This reduces the need for manual hyperparameter sweeps to achieve some compression target: a single run is enough.

## 4.4 PUTTING IT ALL TOGETHER: BALF

Together, these pieces form BALF. An overview can be found in Algorithm 1. The routine NEWLAYER generates a new layer (consisting of two sequential sub-layers) when given the factorized matrices. In practice, we do not select a manual budget $C_{\max}$, but rather a ratio (e.g., we might want to obtain a model with 50% of its original FLOPs). Additionally, note that we retain the original layer if compressing it to the selected rank is not computationally advantageous. There is a small difference between our use of whitening matrices and that of SVD-LLM. We precompute activations, whitening matrices, and factors once; they are then cached. This is significantly more efficient when generating multiple versions of the same model at different compression ratios (see Appendix G for more details about the time cost of each step). In contrast, after compressing a layer, SVD-LLM recomputes that layer's outputs and uses them to compute the whitening matrix of the next layer; this prevents re-use between compression runs. In early experiments, we found that recomputing the outputs offered no significant advantage.

---

**Algorithm 1** BALF overview. Group notation is omitted for simplicity.

---

**Require:** Pre-trained model $\{\mathcal{M}_l\}_{l=1}^L$, calibration data $\mathcal{D}$, budget $C_{\max}$, cost functions $\{C_l\}_{l=1}^L$
**Ensure:** Compressed model (compressed in place for simplicity)
1: $\{\frac{1}{N_l}\overline{I}_l(\mathbf{X}^{l-1})^T\overline{I}_l(\mathbf{X}^{l-1})\}_{l=1}^L \leftarrow \text{COLLECTACTIVATIONS}(\{\mathcal{M}_l\}_{l=1}^L, \mathcal{D})$        ▷ stored in reshaped form
2: $F \leftarrow [\,]$
3: **for** $l = 1, \ldots, L$ **do**
4:     $[\boldsymbol{M}, \boldsymbol{M}^+] \leftarrow \text{BATCHEDUWM}(\frac{1}{N_l}\overline{I}_l(\mathbf{X}^{l-1})^T\overline{I}_l(\mathbf{X}^{l-1}))$; $\mathbf{W} \leftarrow \mathcal{M}_l.\,\text{weights}$
5:     $[\boldsymbol{U}, \boldsymbol{\Sigma}, \boldsymbol{V}^T] \leftarrow \text{BATCHEDSVD}(\boldsymbol{M}^+\overline{P}(\mathbf{W}))$
6:     $F \leftarrow F \cup [\boldsymbol{U}, \boldsymbol{\Sigma}, \boldsymbol{V}^T, \boldsymbol{M}, \boldsymbol{M}^+]$
7: **end for**
8: $\{P_l\}_{l=1}^L \leftarrow \text{ALLOCATERANKS}(\text{GETSIGMAS}(F), C_{\max}, \{C_l\}_{l=1}^L)$
9: **for** $l = 1, \ldots, L$ **do**
10:     **if** $C_l(P_l) \geq \text{ORIGINALCOST}(l)$ **then continue**        ▷ factorizing might not be worth it
11:     $[\boldsymbol{U}, \boldsymbol{\Sigma}, \boldsymbol{V}^T, \boldsymbol{M}, \boldsymbol{M}^+] \leftarrow F[l]$; $\boldsymbol{W}_0 \leftarrow \boldsymbol{M}\boldsymbol{U}\boldsymbol{\Sigma}^{1/2}$; $\boldsymbol{W}_1 \leftarrow \boldsymbol{\Sigma}^{1/2}\boldsymbol{V}^T$
12:     $\mathcal{M}_l \leftarrow \text{NEWLAYER}(\boldsymbol{W}_0, \boldsymbol{W}_1, \mathcal{M}_l.\,\text{bias}; P_l)$        ▷ truncates as needed
13: **end for**

---

**Handling complex network structures.** We note that our method operates at the layer level (it modifies the internal computation of each layer but does not alter the connections between them). Consequently, it automatically handles any complex structures (like residual connections) without requiring any manual input.

**Composing BALF with other compression frameworks.** A useful feature of low-rank factorization is that it can be readily composed with other compression methods. Since each layer is replaced by two sequential layers of the same class, one can directly apply other compression techniques (such as quantization or pruning) to the resulting layers, without any additional adaptation.

**Bounding the output distortion of a model.** In general, providing bounds on the cumulative model distortion incurred when compressing it is challenging. Interestingly, talking in terms of $\ell^{\text{activ}}$ (see Equation (3)) allows for tighter bounds. In Theorem 3, we prove a network-level bound on the output distortion of sequential networks.

**Theorem 3.** *(Informal) Define a sequential L-layer network*

$$\mathbf{X}^0 = \mathbf{X}, \quad \mathbf{X}^l = a_l\left(f_l(\mathbf{X}^{l-1}; \mathbf{W}^l) + \boldsymbol{b}^l\right) = a_l\left(\overline{O}_l(\overline{I}_l(\mathbf{X}^{l-1})\overline{P}_l(\mathbf{W}^l)) + \boldsymbol{b}^l\right),$$

*where $a_l$ is an element-wise activation function with Lipschitz constant $A_l$. Moreover, let $B_l$ be the Lipschitz constant of $\overline{I}_l$ (recall that $\overline{I}_l$ is linear). We assume all layers are ungrouped. Then,*

$$\frac{1}{\sqrt{B}}\big\|\mathbf{X}^L - \widehat{\mathbf{X}}^L\big\|_F \;\leq\; \sum_{l=1}^{L}\Big[A_l\sqrt{\ell_l^{\mathrm{activ}}(P_l)}\prod_{i=l+1}^{L}\|\overline{P}_i(\widehat{\mathbf{W}}^i)\|_2 A_i B_i\Big].$$

A more detailed statement and discussion, along with the proof and special cases, appear in Appendix D. The tightness of the bound depends on the model's depth and its weights, and it is typically loose in practical deep networks. Even so, it remains conceptually informative.

## 5 EXPERIMENTS

This section presents our experimental results, covering a range of architectures, datasets, and scales. Our code is available at https://anonymous.4open.science/r/BALF-2AFC. On CIFAR-10 (Krizhevsky, 2009), we evaluate ResNet-20 and ResNet-56, while on ImageNet (Deng et al., 2009), we consider ResNet-18 and ResNet-50 (He et al., 2016), ResNeXt-50 (32×4d) and ResNeXt-101 (32×8d) with grouped convolutions (Xie et al., 2017), ViT-B/16 (Dosovitskiy et al., 2021), DeiT-B/16 (Touvron et al., 2021), and MobileNet-V2 (Sandler et al., 2018). For all models, every layer is considered for factorization. Calibration is performed using 1024 images for CIFAR-10 and 8192 images for ImageNet, both uniformly sampled from their respective training datasets. The precise settings and hyperparameters are provided in Appendix E.

**Comparison with factorization approaches.** First, we perform an extensive sweep showcasing the complexity–accuracy curves for FLOP and parameter counts. To highlight which components of BALF are important, each run is performed using different methods (all implemented by us): SVD with the energy criterion, activation-aware SVD with the energy criterion, and BALF (activation-aware decomposition with our rank allocator); see Section 4.3 for an overview.

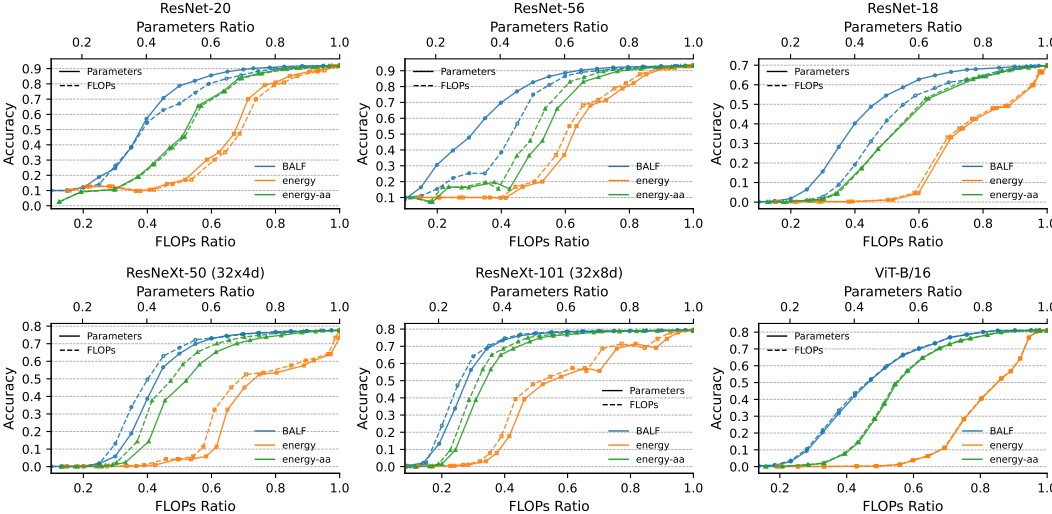

Figure 2: Sweep over different compression ratios with different decomposition methods on different models. Each figure shows six curves. Three of them represent the parameters–accuracy curves and the other three represent the FLOPs–accuracy curves. BALF is set with parameter count and FLOP count ratio targets in the parameters–accuracy and FLOPs–accuracy curves, respectively.

Figure 2 summarizes the complexity–accuracy trade-offs across the different settings. The qualitative trend is consistent: activation-aware decomposition yields clear gains, with the magnitude of the gap varying by configuration. In contrast, standard SVD degrades accuracy even at modest compression ratios, making it impractical without fine-tuning. Finally, our rank-allocation method enables further low-distortion compression—for example, we can reduce the FLOPs of the ResNeXt-101 model by 50% with only about 1.5 percentage points of top-1 accuracy drop.

**Comparison with other fine-tuning-free methods.** Additionally, we compare our factorization method with other compression methods in the fine-tuning-free regime.

For ResNet-50, we compare BALF with several pruning methods: DFPC (Narshana et al., 2023), IterTVSPrune (ITVSP) (Murti et al., 2023), and IFM (Chen et al., 2024). For ResNet-20, we compare against the pruning approach FVRCP (He et al., 2020) and the factorization method ALDS (Liebenwein et al., 2021).

For DeiT-B/16, we compare our method with two types of techniques: GTP (Xu et al., 2024), a token-merging approach that reduces inference FLOPs (but not model size), and two pruning methods: DC-ViT (Zhang et al., 2024) and PRACTISE (Wang & Wu, 2023) (using the results reported by Zhang et al. (2024)).

Compressing compact models such as MobileNet-V2 is known to be challenging. For this model, we compare against the factorization methods SVD-NAS (Yu & Bouganis, 2023), ALDS (Liebenwein et al., 2021), and LR-S2 (Idelbayev & Carreira-Perpiñán, 2020) (using the no-fine-tuning results for ALDS and LR-S2 reported by Yu & Bouganis (2023)). We also compare with the pruning-restoration method proposed by Lee et al. (2025), using their L2 variant (denoted L2+REST). Note that SVD-NAS requires an expensive search procedure (on the order of hours), and LR-S2 allocates ranks through a training run, making them substantially more expensive than BALF. Table 1 contains the different results. In general, BALF surpasses or is competitive with all the compared methods.

**BALF is efficient.** On an RTX 2070 laptop, compressing ImageNet models completes in minutes (under four minutes for all models except ResNeXt-101, which takes less than seven). The most expensive steps can be cached, reducing subsequent compression runs to seconds. We provide a comprehensive benchmark and implementation details in Appendix G.

Table 1: Comparison of different methods at different compression configurations. BALF-P-$x$ and BALF-F-$y$ denote our method when set with parameter and FLOP count objectives, respectively. "—" is used for results that were not reported. $\Delta$F% and $\Delta$P% denote the percentage changes in FLOPs and parameter counts relative to the original model.

(a) ResNet-50 on ImageNet

| Method | $\Delta$F% | $\Delta$P% | $\Delta$Top-1 pp |
|---|---|---|---|
| BALF-F-0.5 | −50.04 | −43.86 | −13.84 |
| BALF-P-0.5 | −28.18 | −50.08 | −7.78 |
| BALF-F-0.7 | −30.00 | −21.69 | −3.48 |
| BALF-P-0.7 | −13.55 | −29.94 | −1.96 |
| BALF-F-0.8 | −20.00 | −9.77 | −1.65 |
| BALF-P-0.8 | −9.03 | −21.92 | −1.12 |
| IFM | — | −5.65 | −1.46 |
| IFM | — | −20.42 | −10.45 |
| ITVSP | — | −4.76 | −3.02 |
| ITVSP | — | −9.98 | −10.21 |
| DFPC | — | −5.66 | −5.78 |
| DFPC | — | −10.92 | −13.88 |

(b) ResNet-20 on CIFAR-10

| Method | $\Delta$F% | $\Delta$P% | $\Delta$Top-1 pp |
|---|---|---|---|
| BALF-F-0.5 | −50.02 | −49.48 | −24.89 |
| BALF-P-0.5 | −24.75 | −49.74 | −13.21 |
| BALF-F-0.7 | −30.74 | −29.10 | −6.26 |
| BALF-P-0.7 | −12.77 | −29.88 | −2.50 |
| BALF-F-0.8 | −20.03 | −17.90 | −2.16 |
| BALF-P-0.8 | −9.27 | −21.70 | −1.20 |
| ALDS | — | −23.71 | −1.50 |
| ALDS | — | −30.05 | −3.37 |
| ALDS | — | −50.00 | −25.22 |
| FVRCP | −19.84 | — | −2.5 |
| FVRCP | −29.51 | — | −5.83 |
| FVRCP | −49.35 | — | −24.86 |

(c) DeiT-B/16 on ImageNet

| Method | $\Delta$F% | $\Delta$P% | $\Delta$Top-1 pp |
|---|---|---|---|
| BALF-F-0.6 | −38.37 | −39.52 | −3.88 |
| BALF-P-0.6 | −38.71 | −39.87 | −4.11 |
| BALF-F-0.7 | −28.78 | −29.64 | −1.08 |
| BALF-P-0.7 | −29.04 | −29.91 | −1.09 |
| BALF-F-0.8 | −19.21 | −19.79 | −0.14 |
| BALF-P-0.8 | −19.36 | −19.94 | −0.16 |
| GTP-15.3 | −13.07 | 0.00 | 0.00 |
| GTP-8.8 | −50.00 | 0.00 | −3.50 |
| PRACTISE | −16.6 | — | −2.5 |
| DC-ViT | −16.6 | — | −0.54 |

(d) MobileNet-V2 on ImageNet

| Method | $\Delta$F% | $\Delta$P% | $\Delta$Top-1 pp |
|---|---|---|---|
| BALF-P-0.7 | −2.06 | −29.70 | −27.65 |
| BALF-P-0.75 | −1.11 | −24.79 | −6.13 |
| BALF-P-0.8 | −0.23 | −19.80 | −0.59 |
| BALF-P-0.9 | −0.13 | −10.76 | −0.25 |
| BALF-F-0.97 | −2.51 | −1.55 | −4.95 |
| SVD-NAS | −15.09 | −9.00 | −12.54 |
| ALDS | −2.62 | −37.61 | −16.95 |
| LR-S2 | −3.81 | −6.24 | −17.46 |
| L2+REST | — | −5.00 | −4.42 |
| L2+REST | — | −10.00 | −18.47 |

**Other results.** On CIFAR-10-C (Hendrycks & Dietterich, 2019), BALF remains robust to distribution shift: when whitening matrices were estimated on the standard CIFAR-10 training set and evaluation was performed on CIFAR-10-C, the additional accuracy drop was typically less than 2.5 pp. Moreover, we found that a smaller number of calibration samples (about 256 and 1024 for CIFAR-10 and ImageNet, respectively) often yields only marginally worse results than those reported in this section. For instance, see Figure 3 for a sweep on ResNet-50. The remaining results can be found in Appendix F.

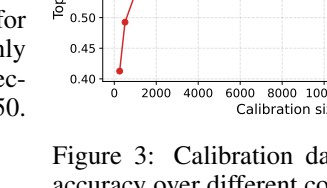

Figure 3: Calibration dataset size vs. accuracy over different compression ratios (in terms of parameter counts) on ResNet-50.

**Limitations and future work.** Even when surpassing (or being competitive with) other methods, compressing compact models (like MobileNet-V2) remains difficult. End-to-end speedups are notable, but we believe that implementing specialized operators would yield better gains (see Appendix F.3 for a discussion). We speculate that combining BALF with expensive search methods (e.g., SVD-NAS) could yield excellent results at the expense of compression time.

## 6 CONCLUSION

In this work, we present BALF, a principled and general activation-aware factorization framework with a budgeted rank allocator that lets users specify FLOPs or parameter count targets directly. Across diverse architectures and scales, BALF delivers strong accuracy-efficiency trade-offs—often surpassing existing fine-tuning-free approaches—while achieving low factorization time on commodity GPUs. This positions BALF as a practical path to model compression without costly search or fine-tuning.

### REPRODUCIBILITY STATEMENT

We have taken several steps to ensure that our results are reproducible. Appendix E provides a detailed description of our experimental setup, and we make our code publicly available at https://anonymous.4open.science/r/BALF-2AFC. The repository includes algorithm implementations, experiment and launch scripts with the exact parameters used, and fixed seeds for every run. It also provides installation instructions and a complete list of dependencies.

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

## APPENDIX OVERVIEW

This appendix is organized as follows:

- Appendix A expands on the factorization scheme and provides parameter and FLOP counts for layers before and after low-rank factorization. It includes results for fully connected, ungrouped convolutional, and grouped convolutional layers, and motivates the use of a shared per-group rank for the latter.

- Appendix B details the construction of uncentered whitening matrices via eigendecomposition, discusses numerical choices and fallbacks, and describes the incremental estimation of second moments and other details.

- Appendix C discusses the specifics of our rank allocator: it specifies the cost model, and presents the previously mentioned Lagrangian relaxation, together with complexity analysis and experimental results on its behavior.

- Appendix D states and proves the formal counterpart of Theorem 3 (discussed in the main text), including simplified forms for special cases.

- Appendix E documents the experimental details of our main experiments, including datasets, models, preprocessing, compression protocols, and other important details.

- Appendix F includes additional experiments that answer secondary questions, including reports on calibration dataset size ablations, robustness under distribution shift (on CIFAR-10-C), practical speedups, and additional accuracy-complexity curves that did not appear in the main text due to space constraints.

- Appendix G describes the end-to-end practical implementation of BALF at a high level and provides timing and peak memory measurements of its different parts.

- Appendix H provides a brief analysis of BALF through the lens of information geometry, following the work of Shumaylov et al. (2025).

- Appendix I collects complete proofs for results deferred from the main text, namely Theorems 1 and 2 and auxiliary lemmas.

- Appendix J discloses the role of LLMs in this work.

## A  DECOMPOSING THE LAYERS AND COMPLEXITY AFTER FACTORIZATION

In this section, we provide a more thorough discussion of how layers are decomposed. We also provide the parameter counts and FLOP counts before and after decomposition (the latter is used as the cost function in our rank allocator). Given a layer and a rank $P$, our scheme has the same complexity as basic SVD truncation to rank $P$; the difference lies in the (expected) quality of the outputs of the factorized layer. We omit biases for simplicity, as these remain constant in the original and factorized layers. $B$ denotes the batch size. Strictly speaking, the FLOP counts reported below should all be doubled (as it stands, we count each multiply-accumulate as one FLOP), but we omit the doubling factor for readability.

### A.1  FULLY CONNECTED LAYERS

Let the weight be $\boldsymbol{W} \in \mathbb{R}^{D_i \times D_o}$. For $\boldsymbol{X} \in \mathbb{R}^{B \times D_i}$, the original layer computes

$$f(\boldsymbol{X}; \boldsymbol{W}) = \boldsymbol{X}\boldsymbol{W}.$$

It has a total of $D_i D_o$ parameters, and takes $B D_i D_o$ FLOPs.

**After decomposition.** Given a rank $P$, we factor the weights into $\boldsymbol{W}_0 \in \mathbb{R}^{D_i \times P}$ and $\boldsymbol{W}_1 \in \mathbb{R}^{P \times D_o}$. The decomposed layer becomes

$$f(\boldsymbol{X}; \boldsymbol{W}_0, \boldsymbol{W}_1) = (\boldsymbol{X}\boldsymbol{W}_0)\boldsymbol{W}_1.$$

The total storage needed is the sum of the storage for each matrix, that is, $P(D_i + D_o)$, while the number of FLOPs is $BP(D_i + D_o)$ (the computation order is algebraically irrelevant, but the order is important for efficiency).

### A.2  2D CONVOLUTIONAL LAYERS

Let the kernel be $\mathbf{W} \in \mathbb{R}^{C_o \times C_i \times H_k \times W_k}$ and the (per-item) output spatial size be $H_o \times W_o$. For an input $\mathbf{X} \in \mathbb{R}^{B \times C_i \times H_i \times W_i}$, the original layer computes

$$f(\mathbf{X}; \mathbf{W}) = \mathbf{X} * \mathbf{W},$$

where $*$ denotes 2D convolution. The total number of parameters is $C_o C_i H_k W_k$, and the number of FLOPs is $B H_o W_o C_o C_i H_k W_k$.

**After decomposition.** Choose a rank $P$. We factor the kernel into

$$\mathbf{W}_0 \in \mathbb{R}^{P \times C_i \times H_k \times W_k} \quad \text{and} \quad \mathbf{W}_1 \in \mathbb{R}^{C_o \times P \times 1 \times 1},$$

and the layer computes

$$f(\mathbf{X}; \mathbf{W}_0, \mathbf{W}_1) = (\mathbf{X} * \mathbf{W}_0) * \mathbf{W}_1.$$

The total number of parameters is $P(C_i H_k W_k + C_o)$, and the FLOPs are $B H_o W_o P(C_i H_k W_k + C_o)$. Note that $H_o$ and $W_o$ subsume the additional convolution hyperparameters (padding, strides, dilation, etc.).

"Regular" convolutions are a special case of grouped convolutions, but we included a separate discussion for the convenience of the reader. We will now discuss the general case.

### A.3  GROUPED 2D CONVOLUTIONAL LAYERS

Let $G$ be the number of groups (assume $C_i$ and $C_o$ are divisible by $G$, as is generally required in modern frameworks). The grouped kernel has shape

$$\mathbf{W} \in \mathbb{R}^{C_o \times C_i/G \times H_k \times W_k}.$$

The original grouped convolution computes

$$f(\mathbf{X}; \mathbf{W}) = \mathbf{X} *_G \mathbf{W},$$

with a total of $C_o \frac{C_i}{G} H_k W_k$ parameters, and a total of $B H_o W_o C_o \frac{C_i}{G} H_k W_k$ FLOPs.

**After decomposition.** Given the per-group rank $P$, we decompose the weights into

$$\mathbf{W}_0 \in \mathbb{R}^{PG \times C_i/G \times H_k \times W_k} \quad \text{and} \quad \mathbf{W}_1 \in \mathbb{R}^{C_o \times P \times 1 \times 1},$$

and the layer computes two grouped convolutions in sequence:

$$f(\mathbf{X}; \mathbf{W}_0, \mathbf{W}_1) = (\mathbf{X} *_G \mathbf{W}_0) *_G \mathbf{W}_1.$$

The total parameter storage across all groups is $P(C_i H_k W_k + C_o)$, and the FLOPs are $B H_o W_o P(C_i H_k W_k + C_o)$.

The intermediate feature map has $PG$ channels in total—$P$ per group—but the parameter and FLOP counts simplify to the expressions above.

### A.3.1 ON THE SHARING OF GROUP RANKS

As noted in the preceding analysis and in the main text, each group shares the same rank $P$. This is done for two main reasons.

First, as noted above, the decomposition with the same rank per group results in two sequential (grouped) convolutions. Having different ranks per group would mean that, implementation-wise, we would need $G$ separate convolutions (or at least some kind of bucketing scheme). In practice, this would be very challenging to perform efficiently.

The second reason is that, even if we managed to achieve an optimal implementation, grouped convolutions are, in essence, $G$ smaller convolutions run in parallel. It is expected that the largest group would generally be a bottleneck for the others, as the smaller groups would have to wait for them.

## B OBTAINING THE UNCENTERED WHITENING MATRICES

In this section, we provide more details about how the uncentered whitening matrices are obtained. We first begin with a discussion of past work.

The naive option to obtain an uncentered whitening matrix is to use the Cholesky decomposition of the second moments, as explained in (Wang et al., 2025c). An issue is that we cannot compute it in the case of singular positive-semidefinite matrices. In their implementation[2], they address this issue by perturbing the matrix with random noise. Later, in Wang et al. (2025b), the authors note this issue and propose the use of an SVD-based whitening matrix, which supports data residing in a subspace (although their analysis does not seem to cover this case). Nevertheless, they show slight end-to-end accuracy gains coming from the SVD-based whitening.

As noted in Section 4.2, we use an eigendecomposition-based scheme, implemented in practice through the eigh routine in PyTorch (a routine that obtains the eigenvalues and eigenvectors of a Hermitian (symmetric in the case of $\mathbb{R}$) matrix, and is usually faster than calling the corresponding generic eigendecomposition routine). First, we show that the eigendecomposition scheme is a valid approach.

Recall that, for a single group $g$, with $\boldsymbol{X} = \overline{I}(\mathbf{X})_g \in \mathbb{R}^{N \times I}$, we compute

$$[\boldsymbol{V}, \boldsymbol{\Lambda}] = \mathsf{eigh}(\frac{1}{N}\boldsymbol{X}^T\boldsymbol{X}),$$

and set

$$\boldsymbol{M} = \boldsymbol{V}\boldsymbol{\Lambda}^{+1/2}, \quad \boldsymbol{M}^+ = \boldsymbol{\Lambda}^{1/2}\boldsymbol{V}^T,$$

where

$$\boldsymbol{V}\boldsymbol{\Lambda}\boldsymbol{V}^T = \frac{1}{N}\boldsymbol{X}^T\boldsymbol{X}.$$

It is easy to show that it satisfies our definition. It follows that

$$\boldsymbol{\Lambda}^{+1/2}\boldsymbol{V}^T\boldsymbol{X}^T\boldsymbol{X}\boldsymbol{V}\boldsymbol{\Lambda}^{+1/2} = N\boldsymbol{\Lambda}^{+1/2}\boldsymbol{V}^T\boldsymbol{V}\boldsymbol{\Lambda}\boldsymbol{V}^T\boldsymbol{V}\boldsymbol{\Lambda}^{+1/2} = N\overline{\boldsymbol{I}}_R,$$

$R$ being the rank of $\boldsymbol{X}$. Moreover, it is easy to see that $\mathrm{row}(\boldsymbol{X}) = \mathrm{range}(\boldsymbol{M})$, as required in our definition (see Section 3). Additionally, the pseudoinverse of $\boldsymbol{\Lambda}$ (a diagonal matrix) can be obtained efficiently by inverting each nonzero element of the diagonal, while leaving zeros as is. This avoids additional numerical approximation errors when forming $\boldsymbol{M}$. We do not perform any additional steps on the whitening matrices.

**Practical implementation.** We use the linear algebra functions (with CUDA tensors) provided in the PyTorch library. During early experimentation, we used the default solver (cuSOLVER (NVIDIA, 2025)), but we noticed that it sometimes crashed for seemingly well-behaved matrices. Because of that, we switched to the MAGMA (Heroux et al., 2024) backend, which behaved much better. Sporadically, the eigh routine was still unable to converge, even when the matrix was actually symmetric. For that reason, we added a fallback to the regular eig routine. We observed that the

---

[2]https://github.com/AIoT-MLSys-Lab/SVD-LLM

fallback was needed only for a couple of matrices, and only in the ResNeXt-101 model. In none of our experiments did the fallback also fail (so we do not resort to noise addition in any instance, contrasting with Wang et al. (2025c)).

**The alternative.** An SVD-based UWM, as proposed in SVD-LLM V2 (Wang et al., 2025b), can be obtained in a similar way with the SVD of $\boldsymbol{X}^T\boldsymbol{X}$. Although it is numerically equivalent, it tends to be slower than our approach.

**Accumulating the uncentered moments incrementally.** The naive way of obtaining $\frac{1}{N}\boldsymbol{X}^T\boldsymbol{X}$ in practice is to use a single batch, and then cache activations by passing the batch through the network. However, for some networks with high-dimensional features and/or large batch sizes, the memory requirements might make the process expensive or impossible. A nice property (mentioned in Wang et al. (2025c), but not discussed extensively) is that we can use data batches to accumulate the second moments. Suppose we have $M$ batches, then,

$$
\frac{1}{N}\boldsymbol{X}^T\boldsymbol{X} = \frac{1}{N}\begin{bmatrix}\boldsymbol{X}_{1,:}^T \\ \vdots \\ \boldsymbol{X}_{M,:}^T\end{bmatrix}[\boldsymbol{X}_{1,:}\cdots\boldsymbol{X}_{M,:}] = \frac{1}{N}\left(\boldsymbol{X}_1^T\boldsymbol{X}_1 + \cdots + \boldsymbol{X}_M^T\boldsymbol{X}_M\right),
$$

where $N$ is the total outer dimension after $\overline{I}$ reshapes the activations (e.g., for convolutions, $N = BH_oW_o$). Here, the indexing denotes the selection of a sub-batch. We use this approach in our implementation.

## C  RANK ALLOCATION DETAILS

This section provides details about the rank allocation strategy discussed in Section 4.3.

Our objective is to select a list $\{P_l\}_{l=1}^L$ denoting the retained rank in each layer, with the objective of maximizing the global retained energy under some cost constraint. We can model this as a binary linear program in the form of a multiple-choice knapsack problem (for a review, see Szkaliczki (2025)), with decision variables $\{x_{l,t}\}_{l,t=1}^{L,U_l}$, where $x_{l,t} \in \{0,1\}$ denotes whether, at layer $l$, $t$ is selected as the maximum retained rank, and $U_l$ denotes an upper bound on the rank (in our case, the number of singular values). Furthermore, by establishing a function $C_l(P_l)$ that returns the computational cost (e.g., in FLOPs or number of parameters) of keeping up to rank $P_l$ at layer $l$, we can ensure that the complexity of the solution is constrained to some user-set quantity. The binary program can be expressed as:

$$
\begin{aligned}
\max_{\{x_{l,t}\}} \quad & \sum_{l=1}^L\sum_{t=1}^{U_l} E_l(t)\,x_{l,t} \\
\text{s.t.} \quad & \sum_{t=1}^{U_l} x_{l,t} = 1, && l = 1,\ldots,L, \\
& \sum_{l=1}^L\sum_{t=1}^{U_l} C_l(t)\,x_{l,t} \le C_{\max}, \\
& x_{l,t} \in \{0,1\}, && \forall\,l,t.
\end{aligned}
$$

**Are there other options for the objective function?** Recall that we use the total retained energy, which is normalized, as the objective function. Following the insights of Theorem 3 and Theorem 4 (although they are not tight for deeper networks), one might speculate that, with appropriate accounting of each layer's importance, we could achieve better results. During our initial experiments, we also evaluated unnormalized distortion as a metric in place of energy and explored layer-wise weightings based on statistics such as output variance and entropy. None of these alternatives improved performance over the method presented here; instead, they increased complexity and posed additional computational overhead and implementation challenges.

**Measuring the cost.** Our allocator makes use of a cost function. We refer the reader to Appendix A for the discussion on the cost of regular and factorized layers, both in terms of FLOPs and in terms of the number of parameters. It is clear that not all $P_l$ selections result in fewer parameters or FLOPs. Given $P_l$ and a criterion, we decompose the layer only when we achieve gains in terms of that criterion (e.g., a reduction in FLOPs or parameters); see Algorithm 1. To reflect this in the cost function, we define it as a piecewise linear function, where the first region is monotonically increasing and represents the region where decomposing the layer results in compression, and the second remains constant, denoting that the action of "keeping rank $P_l$" will have the cost of the original layer. Summarizing, we define

$$C_l(P_l) = \min\{C_l^{\mathrm{orig}}, C_l^{\mathrm{decomp}}(P_l)\}.$$

We emphasize that this step is important so as to inform the solver that we will not factorize a layer if it is not beneficial.

### C.1 A SCALABLE LAGRANGIAN RELAXATION

The aforementioned program can be solved with publicly available libraries (e.g., PuLP if using Python (Dunning et al., 2011)). The issue is that solving it exactly is known to be NP-hard (see Szkaliczki (2025)). To overcome this, we employ a Lagrangian relaxation. Pseudocode defining our implementation is given in Algorithm 2. It accepts a hyperparameter $I$ denoting the number of iterations to use.

**Complexity.** Each of the $I$ outer iterations scans all the candidates once, giving time $O(I \sum_l U_l)$. The initial doubling to find $\lambda_{\max}$ adds an additional $O(D \sum_l U_l)$ factor. Thus, the total time complexity is $O((I + D + 1) \sum_l U_l)$ (in the case of a nontrivial solution).

**Quality of the solutions.** The algorithm yields an approximate solution, but in every experiment we carried out, it produced good results and saturated (approximately) the target budget.

Put simply, the algorithm aims to search for the ideal balance between penalizing the cost and maximizing energy via a parameter $\lambda$; ideal meaning the one that will retain the most energy yet yield a feasible solution. It starts with an interval where it is guaranteed that an optimal $\lambda$ resides (this is guaranteed because of the nature of our optimization problem, where costs and energies are non-decreasing and positive). That interval is then halved every iteration. Its length decreases exponentially with the number of iterations. The final output need not coincide with the actual ("primal") optimum, but it performs well in practice.

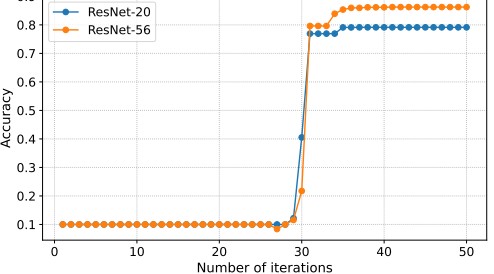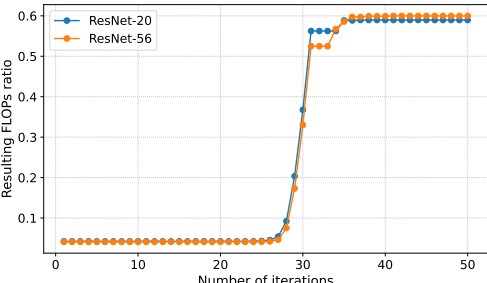

Figure 4: The effect of the number of iterations on CIFAR-10 models. We fix a compression target of 60% of the original FLOPs. The horizontal axis denotes the number of iterations. The vertical axes denote the accuracy achieved (left) and the number of used FLOPs (right). As one can see, the results stabilize with a relatively small number of iterations.

It is, nevertheless, interesting to visualize how the algorithm behaves w.r.t. the number of iterations. Figure 4 compares the number of iterations with two quantities of interest: model accuracy and the actual FLOPs used by the model (when using it with the objective of keeping 60% of the FLOPs).

**Algorithm 2** Lagrangian relaxation for our rank allocator. In practice, he cost and energy functions are precomputed and represented as vectors.

---

**Require:** $L$ layers, each with $U_l$ choices;
 1: energy function $E_l(\cdot)$ and costs $C_l(\cdot)$; total budget $C_{\max}$;
 2: max iterations $I$ (e.g., $I = 300$ in our experiments);
 3: Assumptions (in our problem instance): (1) the problem is feasible (at least one selection with total cost $\leq C_{\max}$ exists); (2) all costs and energies are nonnegative; (3) within each layer $l$, costs and energies are nondecreasing.
**Ensure:** Selection $t_l$ for each layer $l$, one index $t_l \in \{1, \ldots, U_l\}$.
 4: initialize dual bounds $\lambda_{\min} \leftarrow 0$
 5: set $\lambda_{\max} \leftarrow 1$
 6: **if** $\lambda_{\min}$ is feasible **then**
 7:     **return** $\{U_l\}$                                  ▷ this means we can keep everything, it is a trivial solution
 8: **end if**
 9: **repeat**                                                 ▷ grow $\lambda_{\max}$ until feasibility at $\lambda_{\max}$
10:     **for** each layer $l = 1, \ldots, L$ **do**
11:         $t_l \leftarrow \min\big(\arg\max_t \big(E_l(t) - \lambda_{\max} C_l(t)\big)\big)$               ▷ break ties with smallest cost
12:     **end for**
13:     $C \leftarrow \sum_l C_l(t_l)$
14:     **if** $C > C_{\max}$ **then**
15:         $\lambda_{\max} \leftarrow 2\lambda_{\max}$
16:     **end if**
17: **until** $C \leq C_{\max}$                                ▷ now $\lambda_{\max}$ yields a feasible selection
18: **for** $k = 1$ **to** $I$ **do**                          ▷ bisection to shrink the feasible/infeasible bracket
19:     $\lambda \leftarrow (\lambda_{\min} + \lambda_{\max})/2$
20:     **for** each layer $l = 1, \ldots, L$ **do**
21:         $t_l \leftarrow \min\big(\arg\max_t \big(E_l(t) - \lambda C_l(t)\big)\big)$               ▷ break ties with smallest cost
22:     **end for**
23:     $C \leftarrow \sum_l C_l(t_l)$
24:     **if** $C > C_{\max}$ **then**
25:         $\lambda_{\min} \leftarrow \lambda$
26:     **else**
27:         $\lambda_{\max} \leftarrow \lambda$                                 ▷ keep smallest feasible $\lambda$
28:     **end if**
29: **end for**                                    ▷ bracket invariant: $C(\lambda_{\min}) > C_{\max}, C(\lambda_{\max}) \leq C_{\max}$
30: **for** each layer $l = 1, \ldots, L$ **do**
31:     $t_l \leftarrow \min\big(\arg\max_t \big(E_l(t) - \lambda_{\max} C_l(t)\big)\big)$
32: **end for**
33: $C \leftarrow \sum_l C_l(t_l)$
34: **return** $\{t_l\}$                                       ▷ final feasible selection, excellent in practice

---

**Implementation details.** We implement the algorithm in Python, as even for larger models, it has a negligible runtime (less than 0.2 seconds in all our experiments, see Appendix G). Moreover, although the energy and cost functions are represented in functional form in the algorithm description, in practice, they are precomputed as vectors for each layer.

## D  BOUNDING THE OUTPUT DISTORTION OF A MODEL

In this section, we provide a more thorough analysis of the claim in Theorem 3. We provide a more complete description and its proof. The result applies to sequential (possibly nonlinear) networks with $(\overline{O}, \overline{I}, \overline{P})$-expressible layers (possibly with biases).

**Theorem 4.** *Define a sequential L-layer network*

$$\mathbf{X}^0 = \mathbf{X}, \quad \mathbf{X}^l = a_l\left(f_l(\mathbf{X}^{l-1}; \mathbf{W}^l) + b^l\right) = a_l\left(\overline{O}_l(\overline{I}_l(\mathbf{X}^{l-1})\overline{P}_l(\mathbf{W}^l)) + b^l\right),$$

*where $a_l$ is an element-wise activation function with Lipschitz constant $A_l$. Moreover, let $B_l$ be the Lipschitz constant of $\overline{I}_l$ (note that $\overline{I}_l$ is linear and thus Lipschitz-continuous). We restrict our anal-*

*ysis to the case of ungrouped layers. When approximating $\mathbf{W}^l$ as a low-rank version, we denote the resulting parameter by $\widehat{\mathbf{W}}^l$, and the corresponding cumulatively distorted intermediate activations as $\widehat{\mathbf{X}}^l$.*

*For each layer l, fix the retained rank and let*

$$\sqrt{\ell_l^{\mathrm{activ}}} = \frac{1}{\sqrt{B}} \big\| f_l(\mathbf{X}^{l-1}; \mathbf{W}^l) - f_l(\mathbf{X}^{l-1}; \widehat{\mathbf{W}}^l) \big\|_F,$$

*which reads as the (non-cumulative) error introduced by perturbing $\mathbf{W}^l \to \widehat{\mathbf{W}}^l$. Then, the total output distortion*

$$d^{(1:L)} = \frac{1}{B} \big\| \mathbf{X}^L - \widehat{\mathbf{X}}^L \big\|_F^2$$

*satisfies the bound*

$$\sqrt{d^{(1:L)}} \leq \sum_{l=1}^{L} \Big[ A_l \sqrt{\ell_l^{\mathrm{activ}}} \prod_{i=l+1}^{L} \|\overline{P}_i(\widehat{\mathbf{W}}^i)\|_2 A_i B_i \Big].$$

*We follow the convention that the empty product is equal to 1.*

*Proof.* For brevity, we set $\boldsymbol{W}^l \equiv \overline{P}_l(\mathbf{W}^l)$ (recall that we assume ungrouped layers).

Note that $\|\overline{O}_l(\cdot)\|_F = \|\cdot\|_F$, as $\overline{O}_l(\cdot)$ is a composition of permutations and reshapes. It is also true that $a_l \circ \overline{O}_l = \overline{O}_l \circ a_l$, as $a_l$ is element-wise.

Then:

$$\sqrt{B d^{(1:L)}} = \|\mathbf{X}^L - \widehat{\mathbf{X}}^L\|_F$$
$$= \|a_L(f_L(\mathbf{X}^{L-1}; \mathbf{W}^L) + \boldsymbol{b}^L) - a_L(f_L(\widehat{\mathbf{X}}^{L-1}; \widehat{\mathbf{W}}^L) + \boldsymbol{b}^L)\|_F$$
$$= \|a_L(\overline{O}_L(\overline{I}_L(\mathbf{X}^{L-1})\boldsymbol{W}^L) + \boldsymbol{b}^L) - a_L(\overline{O}_L(\overline{I}_L(\widehat{\mathbf{X}}^{L-1})\widehat{\boldsymbol{W}}^L) + \boldsymbol{b}^L)\|_F$$
$$\leq A_L \|\overline{O}_L(\overline{I}_L(\mathbf{X}^{L-1})\boldsymbol{W}^L) + \boldsymbol{b}^L - \overline{O}_L(\overline{I}_L(\widehat{\mathbf{X}}^{L-1})\widehat{\boldsymbol{W}}^L) - \boldsymbol{b}^L\|_F$$
$$= A_L \|\overline{I}_L(\mathbf{X}^{L-1})\boldsymbol{W}^L - \overline{I}_L(\widehat{\mathbf{X}}^{L-1})\widehat{\boldsymbol{W}}^L\|_F$$
$$= A_L \|\overline{I}_L(\mathbf{X}^{L-1})\boldsymbol{W}^L - \overline{I}_L(\mathbf{X}^{L-1})\widehat{\boldsymbol{W}}^L + \overline{I}_L(\mathbf{X}^{L-1})\widehat{\boldsymbol{W}}^L - \overline{I}_L(\widehat{\mathbf{X}}^{L-1})\widehat{\boldsymbol{W}}^L\|_F$$
$$\leq A_L \|\overline{I}_L(\mathbf{X}^{L-1})\boldsymbol{W}^L - \overline{I}_L(\mathbf{X}^{L-1})\widehat{\boldsymbol{W}}^L\|_F + A_L \|\overline{I}_L(\mathbf{X}^{L-1})\widehat{\boldsymbol{W}}^L - \overline{I}_L(\widehat{\mathbf{X}}^{L-1})\widehat{\boldsymbol{W}}^L\|_F$$
$$= A_L \sqrt{B \ell_L^{\mathrm{activ}}} + A_L \|\overline{I}_L(\mathbf{X}^{L-1})\widehat{\boldsymbol{W}}^L - \overline{I}_L(\widehat{\mathbf{X}}^{L-1})\widehat{\boldsymbol{W}}^L\|_F$$
$$\leq A_L \sqrt{B \ell_L^{\mathrm{activ}}} + \|\widehat{\boldsymbol{W}}^L\|_2 A_L B_L \|\mathbf{X}^{L-1} - \widehat{\mathbf{X}}^{L-1}\|_F$$
$$= A_L \sqrt{B \ell_L^{\mathrm{activ}}} + A_L B_L \|\widehat{\boldsymbol{W}}^L\|_2 \sqrt{B d^{(1:L-1)}}.$$

By unrolling the recursion and dividing by $\sqrt{B}$, one arrives at the claimed statement.

$\square$

**Some special cases.** There are some interesting instances where the bound reduces to simpler expressions. For example, for fully connected layers, $B_l = 1$. Moreover, for 1-Lipschitz activations, such as ReLU (Nair & Hinton, 2010) or GELU (Hendrycks & Gimpel, 2023), $A_l = 1$. For instance, on MLPs with ReLU or GELU activations, the bound reduces to

$$\sqrt{d^{(1:L)}} \leq \sum_{l=1}^{L} \left[ \sqrt{\ell_l^{\mathrm{activ}}} \prod_{i=l+1}^{L} \|\widehat{\boldsymbol{W}}^i\|_2 \right].$$

**Looseness of the bound.** In deep networks, as noted in the main text, this bound is informative rather than practical, as it tends to become looser with depth; the main reason is that the bound involves products of spectral norms. There are some works, for instance Cisse et al. (2017), that aim to constrain the operator norms of the layers of neural networks. Such constraints would directly tighten our result, but they are not enforced in standard architectures.

## D.1 COVERING MORE COMPLEX CASES

We proved the bound for sequential networks of ungrouped layers. It is possible to prove other bounds for specific cases; however, covering all cases becomes convoluted and negatively impacts readability, which is why we have only included this specific case.

However, we outline the main ideas to produce bounds for other cases. For grouped layers, the main idea is that one can bound the "batched" versions of the expressions.

Moreover, for residual connections (in particular, one-step residual layers), we can proceed as follows. We omit biases, as they vanish (in a similar way to the proven result). We distinguish two specific cases.

**Case 1.** Suppose we do not use convolutions in the residual path (either because the sizes match or because we use a non-trainable approach, for instance). Then, the one-step residual layer can be expressed as
$$r(\mathbf{X}; \mathbf{W}) = a(f(\mathbf{X}; \mathbf{W})) + d(\mathbf{X}) = a(\overline{O}(\overline{I}(\mathbf{X})\overline{P}(\mathbf{W}))) + d(\mathbf{X}),$$
where $d(\mathbf{X})$ denotes the operation on the residual path, which is not trainable (such as the identity function, or a basic downsampling operation), $f(\mathbf{X}; \mathbf{W})$ is the convolutional layer, and $a$ is the activation function.

Notice that, for compressed weights $\widehat{\mathbf{W}}$,
$$r(\mathbf{X}; \mathbf{W}) - r(\mathbf{X}; \widehat{\mathbf{W}}) = a(f(\mathbf{X}; \mathbf{W})) - a(f(\mathbf{X}; \widehat{\mathbf{W}})),$$
hence the strategy to bound this case is the same as our original one (the residual paths vanish).

**Case 2.** Now consider the case where the residual path also contains a convolution that will be (potentially) compressed. That is,
$$r(\mathbf{X}; \mathbf{W}) = a_1(f(\mathbf{X}; \mathbf{W})) + a_2(g(\mathbf{X}; \mathbf{W})),$$
where $f(\mathbf{X}; \mathbf{W})$ is the main convolution and $g(\mathbf{X}; \mathbf{W})$ is in the residual pathway. One can then bound the errors as $\|\text{error}_r\| \leq \|\text{error}_f\| + \|\text{error}_g\|$ by the triangle inequality.

**Remarks.** We have outlined the main ideas that would allow us to extend the result to other architectures. We would then proceed by topologically sorting the network's computation graph (in a similar way to the sequential case, where sorting is trivial) and bounding the accumulation of errors.

# E EXPERIMENTAL DETAILS

This section provides additional experimental details. All experiments were carried out using the PyTorch library. We make our code public at https://anonymous.4open.science/r/BALF-2AFC. To ensure reproducibility, we include (1) the library code that implements our methods, as well as the baselines, (2) the Python scripts used to carry out the experiments, (3) the Bash scripts used to call the Python scripts with the specific hyperparameters used, and (4) the dependencies and versions used.

## E.1 CIFAR-10 EXPERIMENTS

**Models.** Since there are no widely adopted pre-trained CIFAR-10 ResNet model weights, we pretrain the ResNet-20 and ResNet-56 models ourselves, following standard practice. Our CIFAR-10 ResNet implementation is based on the fb.resnet.torch codebase, with the shortcut option set to B: the residual branch uses an identity mapping when the input and output shapes match, and otherwise applies a $1 \times 1$ convolution with the appropriate stride followed by batch normalization to match shapes. We use the default PyTorch parameter initialization for all layers.

We use SGD with momentum 0.9, weight decay $10^{-4}$, and a batch size of 128. The learning rate is initialized to 0.1 and reduced by a factor of 0.1 at epochs 100 and 150. We save the final model checkpoint at epoch 200. For data augmentation, we apply 4-pixel zero-padding followed by a random crop to $32 \times 32$ pixels, a random horizontal flip with a probability of 0.5, and per-channel normalization. Under these settings, the final test accuracies are 91.93% for ResNet-20 and 93.18% for ResNet-56.

**Compression.** In order to obtain the $k$-calibration dataset, we pick $k$ random images uniformly from the training dataset. For all our main experiments, we use 1024 images.

All factorizable layers are considered for compression. For the CIFAR-10 ResNet models, this includes all the convolutional layers, including shortcut layers, and the final fully connected classifier.

**Dataset processing.** We do not use preprocessing in our calibration dataset and test dataset, other than normalization, so as to mimic inference behavior.

### E.2 IMAGENET EXPERIMENTS

**Models.** For all the CNNs, we use the implementation provided by the torchvision library (maintainers & contributors, 2016). We also use their V1 checkpoints for all the models. For the experiments on vision transformers, we use the implementation and weights provided in the timm library (Wightman, 2019).

**Compression.** In order to obtain the $k$-calibration dataset, we pick $k$ random images uniformly from the training dataset. For all our main experiments, we use 8192 images.

All factorizable layers are considered for compression. For the CNNs, this includes all their convolutional layers (including shortcuts) and the linear layers present in the classifier. For the vision transformers, this includes the initial convolutional layer, as well as the linear layers inside attention blocks and MLPs. For the attention blocks, we factorize the merged Q-K-V matrix (implemented in timm as a single linear layer).

**Dataset processing.** Following standard inference practice, we first resize images to $256 \times 256$ (using bilinear interpolation for all models except for the vision transformers, for which we use bicubic interpolation) and then center-crop them to $224 \times 224$. We then normalize them as usual. We do this for both the calibration dataset and the test dataset.

### E.3 GENERAL DETAILS

For all our main experiments, we use 300 iterations for our rank allocator (see Appendix C.1). For all the compression sweeps in our experiments section, we conduct a full sweep over the compression targets. For the experiments with our allocator, we select compression ratios (w.r.t. parameter and FLOP counts) between 0.1 and 1.0 in steps of 0.05; additionally, we add 0.975. For the other methods (energy-based methods), we manually select an appropriate set of hyperparameters for each model so that they cover a suitable spectrum of compression ratios.

We follow standard practice when reporting compression results. For FLOP counts, we account for convolutional and fully connected layers, but not other secondary layers. For parameter counts, we include all network parameters, including those that are not compressed (which are typically negligible). Finally, unless otherwise noted, we do not perform any layer fusion (e.g., BN+convolution fusion).

For some comparison results in the experimental section, we used WebPlotDigitizer[3] to extract numeric values from figures when the values were not reported in tables. We took care to perform the extraction as accurately as possible.

### E.4 COMPUTATIONAL RESOURCES

All the experiments presented in the paper were conducted on a laptop with an RTX 2070 and a server with an RTX 4090 (the latter was used mainly for the ImageNet experiments). We note that, for all the models discussed, our framework works and is efficient even on the laptop, but accuracy evaluation is significantly faster on the RTX 4090. Preliminary experimentation that did not make it into the paper was conducted on a variety of hardware configurations, including an A100 40GB and other RTX GPUs.

---

[3] https://automeris.io/wpd

## F    ADDITIONAL EXPERIMENTAL RESULTS

This section presents some additional experimental results that aim to answer secondary questions. We also include compression–accuracy curves for three models that were not included in the main text due to space constraints.

### F.1    HOW IMPACTFUL IS THE CALIBRATION DATASET SIZE?

Although we used 8192 calibration images for ImageNet and 1024 for CIFAR-10, we noticed that moderate calibration dataset sizes worked well. Here, we carry out a sweep over different calibration dataset sizes and compression ratios with parameter compression targets. We use a ResNet-20 and ResNet-56 on CIFAR-10, and ResNet-18 and ResNet-50 on ImageNet. Figure 5 shows the different results. As shown, CIFAR-10 is not particularly sensitive to the size of the calibration dataset. On the other hand, one can easily visualize the need for bigger calibration dataset sizes on ImageNet. This need increases for larger models (in our case, the ResNet-50), and is exacerbated for higher compression ratios. However, after 1024 images, more images yield only marginally better results.

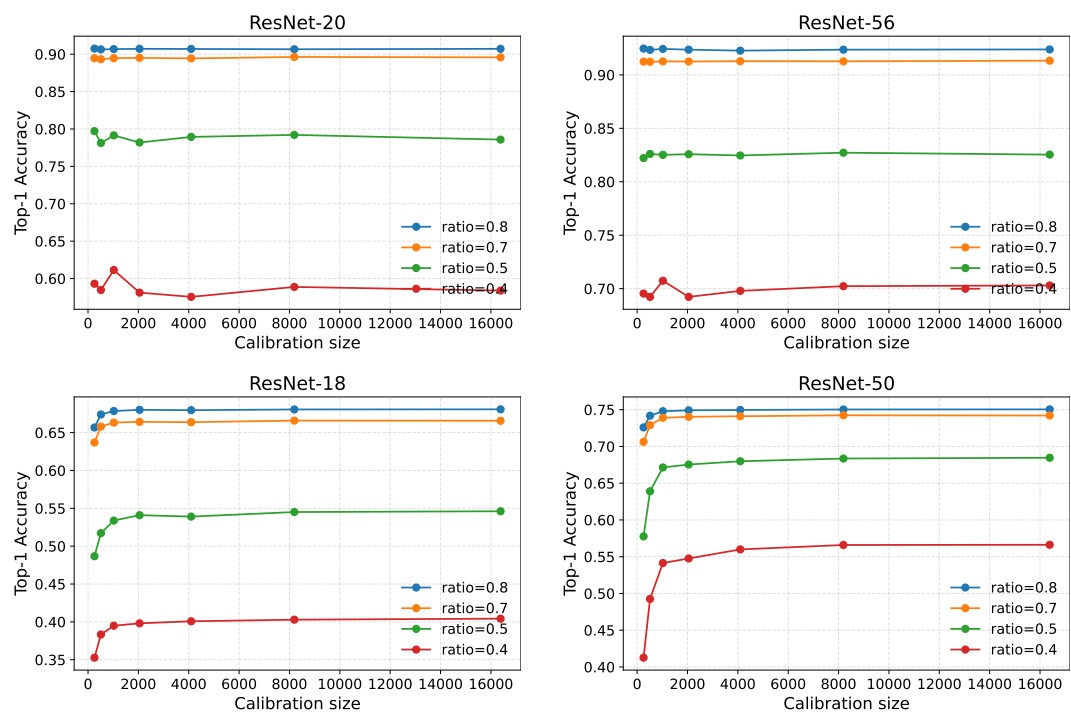

Figure 5: Calibration size vs. performance across models. Hyperparameters are fixed to those of the main experiments, with calibration sizes swept over $256, 512, 1024, 2048, 4096, 8192, 16384$. Ratios indicate the target parameter ratios passed to the allocator.

### F.2    ROBUSTNESS UNDER DISTRIBUTION SHIFT

Our method relies on the computation of (uncentered) whitening matrices (over the training dataset) to estimate the optimal low-rank projections. We rely on the i.i.d. assumption to claim that the statistics of the calibration dataset will be useful for compressing a model in a way that transfers to unseen data. As shown over the training-test pairs on CIFAR-10 and ImageNet, we obtain good results. In general, the i.i.d. assumption is common (although seldom discussed) in methods that use calibration data.

First, we assume that the calibration (a subset of the train dataset, in our case) and the (original) test datasets do satisfy the i.i.d. assumption. Assume we have a third dataset, namely a "corrupted test set", whose generating distribution differs from that of the calibration and test sets.

Given a fixed compression configuration producing a compressed model (with data obtained from the calibration dataset), there is an accuracy gap (on the test dataset) due to the compression procedure, which we denote by $\Delta_{\text{test-comp}}^{\text{test-orig}}$ (comp stands for "compressed", and orig for "original").

Now, if we evaluate our original (non-compressed) model on the corrupted test set, we can measure the accuracy gap as the difference between the accuracy on the original test set and the corrupted test set: $\Delta_{\text{test-corr}}^{\text{test-orig}}$ (corr stands for "corrupted"). For the sake of simplicity, we assume that those are positive (i.e., there are accuracy losses), which is generally true.

We also evaluate our compressed model on the corrupted dataset, and measure $\Delta_{\text{test-comp+corr}}^{\text{test-orig}}$, i.e., the gap between the original test set on our compressed model and the corrupted test set on the compressed model.

We wish to answer the question: How much additional accuracy loss does our compression method induce when evaluated on the corrupted test dataset? Formally, we wish to measure $\Delta_{\text{test-comp+corr}}^{\text{test-orig}} - \Delta_{\text{test-corr}}^{\text{test-orig}}$ (we make the assumption that these are "orthogonal").

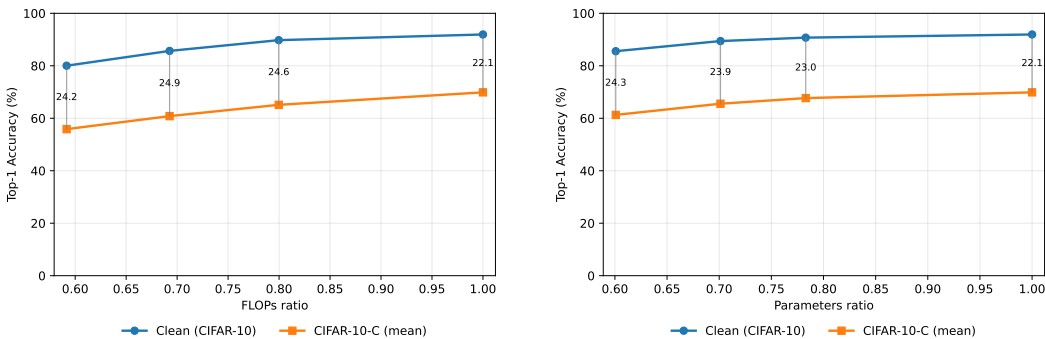

Figure 6: Compression–accuracy curves on ResNet-20 over different compression ratios on CIFAR-10 and CIFAR-10-C as test datasets (the accuracy on CIFAR-10-C is measured as the average accuracy over all corruption types and severities). The left image uses a FLOPs ratio target and the right one uses a parameter count ratio target.

We measure this quantity on the CIFAR-10 dataset. We take our pre-trained ResNet-20 model on CIFAR-10 and compress it across multiple target ratios using BALF (we compress it with different configurations, including FLOPs targets and parameter count targets). We use 1024 CIFAR-10 training images as the calibration set. Additionally, we use the same hyperparameters as in our main experiments. For each resulting compressed model (and for the original model), we then evaluate: (1) its accuracy on the original CIFAR-10 test set, and (2) its accuracy on each corruption and severity level from CIFAR-10-C (Hendrycks & Dietterich, 2019). Over different compression configurations, we report clean accuracy, CIFAR-10-C mean accuracy (over all the corruption configurations), and the gap between them. Finally, we visualize the differences between clean and corrupted accuracies for both models directly and annotate their gaps. Results are shown in Figure 6. This allows us to measure the additional loss in robustness introduced by compression on top of the loss already incurred by distribution shift.

BALF remains surprisingly robust (more than expected, given its direct dependence on the feature distribution), with the compression step causing only small additional drops in accuracy on the corrupted test set.

### F.3 SPEEDUP OF LOW-RANK LAYERS IN PRACTICE

This section discusses the relationship between our compression results (in terms of FLOPs) and the end-to-end speedup obtained in practice. We use basic PyTorch implementations of low-rank operators. In particular, we substitute each original layer with two sequential layers, as described in Appendix A, without any specialized implementation.

We measure the throughput of two ImageNet models (ResNet-50 and ViT-B/16) on an RTX 2070 laptop at different compression ratios. Every batch normalization (BN) (Ioffe & Szegedy, 2015) layer is first removed from the ResNet-50 model so as to mimic the usual inference scenario (where BN layers are folded). We note that our method does not interfere with BN folding in any way.

We found that, in general, compression results translate well to actual speedups (see Figure 7 and Figure 8). However, it is to be noted that convolutional layers fare worse than linear layers in this regard, and that the theoretical gains are not fully exploited (which is, in general, a persistent problem in structured compression methods without specialized operators).

We expect that specialized operator implementations, including those found in software for edge devices, will be able to fully exploit the compression in terms of speedup. In fact, Sui et al. (2024), for instance, were able to obtain near-perfect practical speedups from FLOP reductions on ASICs and FPGAs on ResNet-50 with low-rank layers.

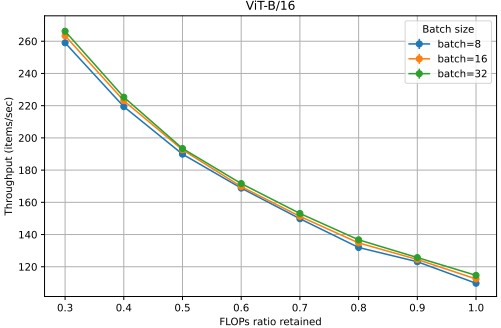 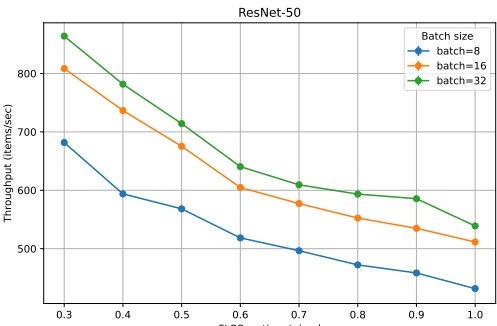

Figure 7: Throughput vs. FLOPs retained on ViT-B/16

Figure 8: Throughput vs. FLOPs retained on ResNet-50

We note that this is, in general, a persistent problem with compression methods (see, e.g., Narshana et al. (2023), where, without specialized implementations, the actual speedups are significantly lower than the FLOPs reductions). We leave the development of specialized operators for future work.

### F.4 ADDITIONAL COMPLEXITY–ACCURACY FIGURES

The main text contained complexity–accuracy figures for six models (although experimental results were provided for all nine), but was missing the ResNet-50, DeiT-B/16, and MobileNet-V2 ones because of space constraints. They are provided in Figure 9.

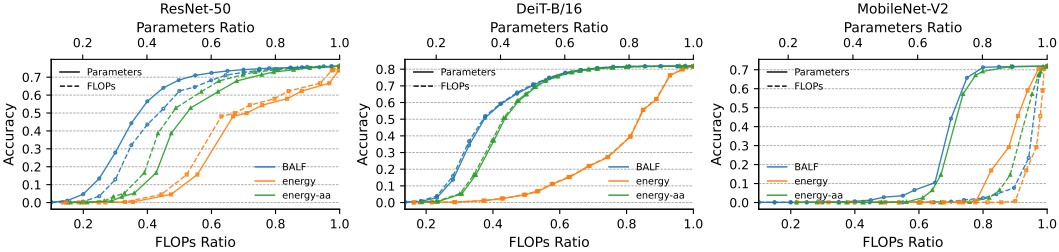

Figure 9: Additional figures with experimental results that did not appear in the main text.

## G RUNTIME AND MEMORY FOOTPRINT OF BALF

Our implementation of BALF is conceptually similar to Algorithm 1, but there are some implementation details that are important to discuss. We will first provide a high-level overview of the implementation details:

1. First, given a calibration dataset, we gather the activation moments. The model runs on the GPU, but the activation moments are offloaded to CPU memory to reduce GPU memory usage and allow compression on lower-end GPUs. Once activation moments are gathered, they can be cached (on disk) for subsequent compression configurations.

2. Then, whitening matrices and factors are computed on the GPU. These are cached on disk to maintain low memory usage and to avoid recomputation in future runs.

3. Our rank allocator uses the singular values (computed in the preceding step) to decide how much to compress each layer.

4. Then, we iterate through the model and substitute the layers with their truncated versions.

Table 2: Runtime breakdown and memory usage for different models. All timings are in seconds. Peak Mem. denotes the peak CUDA memory usage during the run (in GiB).

| Model | Act. | Fact.+Whit. | Solver | Replace | Misc | Total | Peak Mem. |
|---|---|---|---|---|---|---|---|
| ResNet-18 | 78.706 | 12.664 | 0.024 | 3.345 | 0.005 | 94.744 | 1.16 |
| ResNet-50 | 117.995 | 19.565 | 0.069 | 13.765 | 0.003 | 151.396 | 1.34 |
| MobileNet-V2 | 62.911 | 12.758 | 0.051 | 1.061 | 0.002 | 76.783 | 2.23 |
| ResNeXt-50 (32×4d) | 94.983 | 17.089 | 0.060 | 2.906 | 0.002 | 115.040 | 2.29 |
| ResNeXt-101 (32×8d) | 311.523 | 70.968 | 0.124 | 11.382 | 0.005 | 394.001 | 4.45 |
| ViT-B/16 | 188.084 | 34.659 | 0.064 | 7.694 | 0.003 | 230.503 | 0.80 |
| DeiT-B/16 | 183.044 | 35.961 | 0.067 | 18.332 | 0.002 | 237.406 | 0.80 |

For a thorough examination of the implementation, we refer the reader to our code[4]. Table 2 presents the runtime breakdown and peak memory usage of our method on different models on the ImageNet dataset, evaluated on an RTX 2070 laptop. We use 8192 calibration samples (as in our main experiments), divided into batches of 64 samples. All other parameters match those used in the main experiments. We note that the peak memory usage typically occurs during the computation of activation moments, and that using smaller batch sizes can further reduce it.

The only hyperparameters that are expected to affect the runtime of our method are the number of calibration samples and the number of solver iterations. For the first, it is only expected to affect (linearly) the activation-gathering phase (Act. in Table 2).

Second, the number of solver iterations affects (linearly) the time it takes to produce a solution (Solver in Table 2). Its runtime is negligible (less than 0.15 seconds in the runs of this section, using 300 iterations).

Finally, we reiterate that the most expensive steps (activation moment gathering and factorization/whitening) are cached, allowing one to obtain different models with different compression configurations in a matter of seconds on average, even on lower-end hardware, after the first run. We also note that our implementation was not particularly optimized (as compression time is not a bottleneck in our evaluations), and that further runtime gains are probably attainable.

# H A POST-HOC INTERPRETATION VIA INFORMATION GEOMETRY

Recent work by Shumaylov et al. (2025) provides interesting insights into the factorization of neural networks from an information-geometric point of view. BALF has an interesting interpretation through the lens of their framework.

Let $\widehat{\mathbf{W}}$ denote the low-rank approximation of a parameter tensor. They posit that general activation-aware factorization approaches solve an optimization problem equivalent to minimizing the squared output mismatch under an i.i.d. Gaussian noise model with identity covariance. Translating that to

---

[4]https://anonymous.4open.science/r/BALF-2AFC

our framework, we obtain

$$\arg\min_{\widehat{\mathbf{W}}\in\widehat{\mathcal{M}}} \sum_{l=1}^{L} \left\| \overline{I}_l(\mathbf{X}^{l-1})\overline{P}_l(\mathbf{W}^l) - \overline{I}_l(\mathbf{X}^{l-1})\overline{P}_l(\widehat{\mathbf{W}}^l) \right\|_F^2$$

$$= \arg\min_{\widehat{\mathbf{W}}\in\widehat{\mathcal{M}}} \sum_{l=1}^{L}\sum_{g=1}^{G_l} \left\| \overline{I}_l(\mathbf{X}^{l-1})_{g,:,:}\overline{P}_l(\mathbf{W}^l)_{g,:,:} - \overline{I}_l(\mathbf{X}^{l-1})_{g,:,:}\overline{P}_l(\widehat{\mathbf{W}}^l)_{g,:,:} \right\|_F^2$$

$$= \arg\min_{\widehat{\mathbf{W}}\in\widehat{\mathcal{M}}} \sum_{l=1}^{L}\sum_{g=1}^{G_l}\sum_{n=1}^{N_l} \left\| \overline{I}_l(\mathbf{X}^{l-1})_{g,n,:}\overline{P}_l(\mathbf{W}^l)_{g,:,:} - \overline{I}_l(\mathbf{X}^{l-1})_{g,n,:}\overline{P}_l(\widehat{\mathbf{W}}^l)_{g,:,:} \right\|_2^2$$

$$= \arg\min_{\widehat{\mathbf{W}}\in\widehat{\mathcal{M}}} D_{\mathrm{KL}}\big(p_{\mathbf{W}} \,\|\, p_{\widehat{\mathbf{W}}}\big),$$

with

$$p_{\mathbf{W}}(\cdot) = \prod_{l=1}^{L}\prod_{g=1}^{G_l}\prod_{n=1}^{N_l} \mathcal{N}\Big(\cdot \,\Big|\, \overline{I}_l(\mathbf{X}^{l-1})_{g,n}\overline{P}_l(\mathbf{W}^l)_g, \mathbf{I}\Big),$$

and

$$p_{\widehat{\mathbf{W}}}(\cdot) = \prod_{l=1}^{L}\prod_{g=1}^{G_l}\prod_{n=1}^{N_l} \mathcal{N}\Big(\cdot \,\Big|\, \overline{I}_l(\mathbf{X}^{l-1})_{g,n}\overline{P}_l(\widehat{\mathbf{W}}^l)_g, \mathbf{I}\Big).$$

In the preceding expressions, the $n$ index corresponds to the outer dimension (e.g., with $BH_oW_o = N$ for convolutional layers), $g$ to the groups, and $l$ to the layers. The optimization problem generally assumes ranks are fixed ($\widehat{\mathcal{M}}$ denotes a model with layers truncated to fixed ranks, assumed to be given in advance), and it is only concerned with finding the best low-rank projections.

On the other hand, the problem BALF solves can be formulated as

$$\arg\min_{\widehat{\mathbf{W}}\in\widehat{\mathcal{M}}:C(\widehat{\mathcal{M}})\leq C_{\max}} \sum_{l=1}^{L} \frac{\left\| \overline{I}_l(\mathbf{X}^{l-1})\overline{P}_l(\mathbf{W}^l) - \overline{I}_l(\mathbf{X}^{l-1})\overline{P}_l(\widehat{\mathbf{W}}^l) \right\|_F^2}{\left\| \overline{I}_l(\mathbf{X}^{l-1})\overline{P}_l(\mathbf{W}^l) \right\|_F^2}$$

$$= \arg\min_{\widehat{\mathbf{W}}\in\widehat{\mathcal{M}}:C(\widehat{\mathcal{M}})\leq C_{\max}} \sum_{l=1}^{L}\sum_{g=1}^{G_l} \frac{\left\| \overline{I}_l(\mathbf{X}^{l-1})_{g,:,:}\overline{P}_l(\mathbf{W}^l)_{g,:,:} - \overline{I}_l(\mathbf{X}^{l-1})_{g,:,:}\overline{P}_l(\widehat{\mathbf{W}}^l)_{g,:,:} \right\|_F^2}{\left\| \overline{I}_l(\mathbf{X}^{l-1})\overline{P}_l(\mathbf{W}^l) \right\|_F^2}$$

$$= \arg\min_{\widehat{\mathbf{W}}\in\widehat{\mathcal{M}}:C(\widehat{\mathcal{M}})\leq C_{\max}} \sum_{l=1}^{L}\sum_{g=1}^{G_l}\sum_{n=1}^{N_l} \frac{\left\| \overline{I}_l(\mathbf{X}^{l-1})_{g,n,:}\overline{P}_l(\mathbf{W}^l)_{g,:,:} - \overline{I}_l(\mathbf{X}^{l-1})_{g,n,:}\overline{P}_l(\widehat{\mathbf{W}}^l)_{g,:,:} \right\|_2^2}{\left\| \overline{I}_l(\mathbf{X}^{l-1})\overline{P}_l(\mathbf{W}^l) \right\|_F^2}$$

$$= \arg\min_{\widehat{\mathbf{W}}\in C(\widehat{\mathcal{M}}),\, C(\widehat{\mathcal{M}})\leq C_{\max}} D_{\mathrm{KL}}\big(p_{\mathbf{W}} \,\|\, p_{\widehat{\mathbf{W}}}\big),$$

with

$$p_{\mathbf{W}}(\cdot) = \prod_{l=1}^{L}\prod_{g=1}^{G_l}\prod_{n=1}^{N_l} \mathcal{N}\left(\cdot \,\Big|\, \overline{I}_l(\mathbf{X}^{l-1})_{g,n}\overline{P}_l(\mathbf{W}^l)_g, \left\|\big(\overline{I}_l(\mathbf{X}^{l-1})\overline{P}_l(\mathbf{W}^l)\big)\right\|_F^2 \mathbf{I}\right),$$

and

$$p_{\widehat{\mathbf{W}}}(\cdot) = \prod_{l=1}^{L}\prod_{g=1}^{G_l}\prod_{n=1}^{N_l} \mathcal{N}\left(\cdot \,\Big|\, \overline{I}_l(\mathbf{X}^{l-1})_{g,n}\overline{P}_l(\widehat{\mathbf{W}}^l)_g, \left\|\big(\overline{I}_l(\mathbf{X}^{l-1})\overline{P}_l(\mathbf{W}^l)\big)\right\|_F^2 \mathbf{I}\right).$$

There are two main differences. First, BALF also works under the Gaussian assumption, but with the additional nuance that the variances depend on the per-layer output magnitude (in terms of the squared Frobenius norm). Second, the optimization problem is not only concerned with the low-rank projection scheme given a fixed rank per layer; the selection of ranks under some complexity budget is, itself, baked into the optimization problem. (Note that, strictly speaking, BALF solves the problem approximately in practice.)

We believe this point of view might be an interesting avenue for future research.

## I MISSING PROOFS

In this section, we provide full proofs for the results in the main text that were not covered in other sections of the appendix. Results are restated for the convenience of the reader.

### I.1 PROOF OF THEOREM 2

**Theorem 2.** *Assume that $f(\mathbf{X}; \mathbf{W})$ is $(\overline{O}, \overline{I}, \overline{P})$-expressible. Then, the activation distortion (as defined in Equation (3)) incurred when low-rank projecting the parameters with $\mathcal{T}_P^{\mathrm{AA}}(\cdot)$ is $\frac{N}{B} \sum_{g=1}^{G} \sum_{i=P+1}^{U} \boldsymbol{\sigma}_{g,i}^2$, where $\boldsymbol{\sigma}$ denotes the batched singular values of $\mathbf{M}^+ \overline{P}(\mathbf{W})$.*

*Proof.* First, observe that

$$\|\mathsf{cat}(\boldsymbol{A}_1, \dots, \boldsymbol{A}_K)\|_F^2 = \sum_{i=1}^{K} \|\boldsymbol{A}_i\|_F^2, \tag{4}$$

where cat denotes the operation that yields a third-order tensor from a batch of matrices (of the same shape). Therefore, for a batch of distortion matrices, we may analyze each term independently and sum their contributions.

Moreover,

$$\begin{aligned}
\left\| f(\mathbf{X}; \mathbf{W}) - f(\mathbf{X}; \mathcal{T}_P^{\mathrm{AA}}(\mathbf{W})) \right\|_F &= \left\| \overline{O}(\overline{I}(\mathbf{X})\,\overline{P}(\mathbf{W})) - \overline{O}(\overline{I}(\mathbf{X})\,\mathcal{T}_P^{\mathrm{AA}}(\overline{P}(\mathbf{W}))) \right\|_F \\
&= \left\| \overline{I}(\mathbf{X})\,\overline{P}(\mathbf{W}) - \overline{I}(\mathbf{X})\,\mathcal{T}_P^{\mathrm{AA}}(\overline{P}(\mathbf{W})) \right\|_F,
\end{aligned}$$

since $\overline{O}$ is a composition of reshapes and permutes, which preserve the Frobenius norm.

Fix a group index $g$ (omitted from the notation below for simplicity). For reference, the dimensions of the different elements are: $\overline{I}(\mathbf{X}) \in \mathbb{R}^{N \times I}$, $\boldsymbol{M} \in \mathbb{R}^{I \times I}$, $\boldsymbol{M}^+ \in \mathbb{R}^{I \times I}$, and $\overline{P}(\mathbf{W}) \in \mathbb{R}^{I \times O}$.

When using activation-aware factorization and truncating to rank $P$, it follows that

$$\begin{aligned}
\ell_{(g)}^{\mathrm{activ}}(P) &= \frac{1}{B} \left\| \overline{I}(\mathbf{X})\overline{P}(\mathbf{W}) - \overline{I}(\mathbf{X})\boldsymbol{M}\mathcal{T}_P^{\mathrm{SVD}}(\boldsymbol{M}^+\overline{P}(\mathbf{W})) \right\|_F^2 \\
&\underset{\text{(a)}}{=} \frac{1}{B} \left\| \overline{I}(\mathbf{X})\boldsymbol{M}\left(\boldsymbol{M}^+\overline{P}(\mathbf{W}) - \mathcal{T}_P^{\mathrm{SVD}}(\boldsymbol{M}^+\overline{P}(\mathbf{W}))\right) \right\|_F^2 \\
&= \frac{1}{B} \left\| \overline{I}(\mathbf{X})\boldsymbol{M}\boldsymbol{U}\boldsymbol{\Sigma}\boldsymbol{V}^T \right\|_F^2 \\
&= \frac{1}{B} \mathrm{tr}\left(\boldsymbol{V}\boldsymbol{\Sigma}\boldsymbol{U}^T\boldsymbol{M}^T\overline{I}(\mathbf{X})^T\overline{I}(\mathbf{X})\boldsymbol{M}\boldsymbol{U}\boldsymbol{\Sigma}\boldsymbol{V}^T\right) \\
&\underset{\text{(b)}}{=} \frac{N}{B} \mathrm{tr}\left(\boldsymbol{V}\boldsymbol{\Sigma}^2\boldsymbol{V}^T\right) \\
&= \frac{N}{B} \sum_{i=P+1}^{U} \boldsymbol{\sigma}_i^2,
\end{aligned}$$

where the SVD is taken of $\boldsymbol{M}^+\overline{P}(\mathbf{W})$.

For step (a), note that $\overline{I}(\mathbf{X})\boldsymbol{M}\boldsymbol{M}^+ = \overline{I}(\mathbf{X})$ (see Section 3).

For step (b), note the following. First, recall from the definition of $\boldsymbol{M}$ (see Section 3) that $\boldsymbol{M}^T\overline{I}(\mathbf{X})^T\overline{I}(\mathbf{X})\boldsymbol{M} = N\overline{\boldsymbol{I}}_R$ (where $R$ is the rank of $\overline{I}(\mathbf{X})$); and that $\boldsymbol{M}$ has its last $I - R$ columns set to 0, meaning $\boldsymbol{M}^+$ has its last $I - R$ rows set to 0.

This results in $\boldsymbol{U}^T\boldsymbol{M}^T\overline{I}(\mathbf{X})^T\overline{I}(\mathbf{X})\boldsymbol{M}\boldsymbol{U} = N\boldsymbol{U}^T\overline{\boldsymbol{I}}_R\boldsymbol{U} = N\overline{\boldsymbol{I}}_R$ (this last equality follows from the fact that $\boldsymbol{M}^+\overline{P}(\mathbf{W}) \in \mathbb{R}^{I \times O}$ has its last $I - R$ rows set to 0 since $\boldsymbol{M}^+$ does).

From the observation in Equation (4), it follows that the total distortion is the sum across each group's distortion

$$\ell^{\mathrm{activ}}(P) = \sum_{g=1}^{G} \ell_{(g)}^{\mathrm{activ}}(P) = \frac{N}{B} \sum_{g=1}^{G} \sum_{i=P+1}^{U} \boldsymbol{\sigma}_{g,i}^2,$$

which completes the proof.

$\square$

### I.2 PROOF OF THEOREM 1

**Lemma 1.** *Let $\boldsymbol{Y} \in \mathbb{R}^{M \times N}$ satisfy $\boldsymbol{Y}^T\boldsymbol{Y} = \alpha \boldsymbol{I}_N$ with $\alpha > 0$. Then, for any $\boldsymbol{X} \in \mathbb{R}^{N \times K}$ and any $P \geq 1$,*

$$\sum_{i=1}^{P} \boldsymbol{\sigma}_i(\boldsymbol{Y}\boldsymbol{X}) = \sqrt{\alpha} \sum_{i=1}^{P} \boldsymbol{\sigma}_i(\boldsymbol{X}).$$

*Proof.* We have

$$(\boldsymbol{Y}\boldsymbol{X})^T(\boldsymbol{Y}\boldsymbol{X}) = \boldsymbol{X}^T(\boldsymbol{Y}^T\boldsymbol{Y})\boldsymbol{X} = \alpha \boldsymbol{X}^T\boldsymbol{X}.$$

Hence, the (nonzero) eigenvalues of $(\boldsymbol{Y}\boldsymbol{X})^T(\boldsymbol{Y}\boldsymbol{X})$ are exactly $\alpha$ times those of $\boldsymbol{X}^T\boldsymbol{X}$, so $\boldsymbol{\sigma}_i(\boldsymbol{Y}\boldsymbol{X}) = \sqrt{\alpha}\boldsymbol{\sigma}_i(\boldsymbol{X})$ for all $i$. Summing the top $P$ singular values gives the claim. $\square$

**Theorem 1.** $\mathcal{T}_P^{\mathrm{AA}}(\cdot)$ *is optimal in the sense of Definition 2.*

*Proof.* It suffices to prove the result for a single group. Fix a group (its index is omitted from the notation for brevity), and let $\boldsymbol{W} = \overline{P}(\mathbf{W}) \in \mathbb{R}^{I \times O}$, $\boldsymbol{X} = \overline{I}(\mathbf{X}) \in \mathbb{R}^{N \times I}$, $\boldsymbol{M} \in \mathbb{R}^{I \times I}$, and $\boldsymbol{M}^+ \in \mathbb{R}^{I \times I}$ denote the items corresponding to said group, and denote the rank of $\boldsymbol{X}$ by $R$.

We recall the following facts; see Section 3. First, $\boldsymbol{M}$ has its last $I - R$ columns set to $0$, which forces $\boldsymbol{M}^+$ to have its last $I - R$ rows set to $0$. Hence, $\boldsymbol{M}\boldsymbol{M}^+ = \boldsymbol{M}\boldsymbol{I}_{:,:R}\boldsymbol{I}_{:R,:}\boldsymbol{M}^+$. Moreover, from the definition of $\boldsymbol{M}$, we have $\boldsymbol{M}^T\boldsymbol{X}^T\boldsymbol{X}\boldsymbol{M} = N\overline{\boldsymbol{I}}_R$, and therefore, it holds that $\boldsymbol{I}_{:R,:}\boldsymbol{M}^T\boldsymbol{X}^T\boldsymbol{X}\boldsymbol{M}\boldsymbol{I}_{:,:R} = N\boldsymbol{I}_R$. It is also true that $\boldsymbol{X}\boldsymbol{M}\boldsymbol{M}^+ = \boldsymbol{X}$.

It follows that

$$\sum_{i=1}^{P} \boldsymbol{\sigma}_i^2(\boldsymbol{X}\boldsymbol{W}) = \sum_{i=1}^{P} \boldsymbol{\sigma}_i^2(\boldsymbol{X}\boldsymbol{M}\boldsymbol{M}^+\boldsymbol{W}) = \sum_{i=1}^{P} \boldsymbol{\sigma}_i^2(\boldsymbol{X}\boldsymbol{M}\boldsymbol{I}_{:,:R}\boldsymbol{I}_{:R,:}\boldsymbol{M}^+\boldsymbol{W}) = N\sum_{i=1}^{P} \boldsymbol{\sigma}_i^2(\boldsymbol{M}^+\boldsymbol{W}),$$

where the last equality follows from Lemma 1.

Hence,

$$\|\overline{I}(\mathbf{X})\overline{P}(\mathbf{W}) - \mathcal{T}_P^{\mathrm{SVD}}(\overline{I}(\mathbf{X})\overline{P}(\mathbf{W}))\|_F^2 = \sum_{i=P+1}^{U} \boldsymbol{\sigma}_i^2(\boldsymbol{X}\boldsymbol{W})$$

$$= N\sum_{i=P+1}^{U} \boldsymbol{\sigma}_i^2(\boldsymbol{M}^+\boldsymbol{W})$$

$$= \|\overline{I}(\mathbf{X})\overline{P}(\mathbf{W}) - \overline{I}(\mathbf{X})\mathcal{T}_P^{\mathrm{AA}}(\overline{P}(\mathbf{W}))\|_F^2,$$

where the last equality follows from Theorem 2.

Taking square roots completes the proof. $\square$

## J LLM USAGE

In this section, we disclose the role of LLMs in this work.

First, we used LLMs to help identify related work and select comparison baselines (in addition to our manual efforts). The content itself was read by the authors; LLMs primarily assisted with search. We also used them to discuss research ideas and to verify the correctness and coherence of our claims.

LLMs were used to bootstrap, modify, and evaluate code (including checking for correctness). They were particularly useful for iterating on result-visualization scripts (e.g., code that takes experimental outputs and produces plots or table components).

Finally, LLMs were used to evaluate and improve the presentation and writing quality of various drafts (including equation formatting), to suggest missing explanations or experiments, and to assess the overall coherence of the text.

