# OpenReview forum: "BALF: Budgeted Activation-Aware Low-Rank Factorization for Fine-Tuning-Free Model Compression"
_ICLR.cc/2026/Conference — Submitted to ICLR 2026_

### Official Review · Reviewer_LWbM · 2025-10-25

**Soundness:** 3
**Presentation:** 3
**Contribution:** 3
**Rating:** 6
**Confidence:** 4

**Summary:**

This paper introduces BALF, a fine-tuning-free compression pipeline that unifies activation-aware low-rank factorization across fully connected, convolutional, and grouped-convolution layers, and couples it with a lightweight budgeted rank allocator based on Lagrangian relaxation that directly targets user-specified FLOPs or parameter budgets. The method estimates uncentered-whitening transforms from a small calibration set, performs truncated SVD in the whitened domain to minimize expected layer-output distortion, and then selects per-layer ranks globally to satisfy a compute/size constraint, replacing layers with efficient two-stage low-rank modules only when they yield real savings.

**Strengths:**

- General, principled activation-aware factorization applicable to grouped convolutions, not just FC layers; the method uses whitening to minimize expected layer-output distortion.
- Budget-aware rank allocation: sets layer-wise ranks to meet FLOPs or parameter budgets via a Lagrangian relaxation.
- Broad empirical coverage across CNNs and ViTs, with competitive fine-tuning-free trade-offs.

**Weaknesses:**

- Innovation scope: activation-aware factorization is not entirely new; the paper’s novelty lies in generalizing it to conv/grouped-conv and coupling it with an efficient budgeted allocator. This is a strong systems contribution but less of a theoretical breakthrough.
- The model-level bound (Theorem 3 in the main text; formal version in the appendix) is derived for a sequential L-layer network and does not explicitly handle residual/skip connections typical of ResNets. While the empirical section includes ResNet-family models, the paper does not clarify how residual blocks are treated in the theoretical bound (e.g., block-level aggregation or additional terms due to additive skips) nor in the practical error accounting within a residual structure.

**Questions:**

- Can BALF be composed with post-training quantization?
- When applying BALF to ResNet families, do you fuse BN into Conv before factorization, or do you factorize Conv and then correct BN statistics?

---

> ### Author Response · Authors · 2025-11-13
> **Response to Reviewer LWbM**
>
> We thank you for the thoughtful review, the good score, and the attention to detail in your comments. We will do our best to address the different points.
>
> >**W1** Innovation scope
>
> We agree with you that activation-aware factorization is not entirely new. However, we note that we not only generalize it with our $(\overline{O}, \overline{I}, \overline{P})$ framework, but also rework the core theory to make it more robust.
>
> In particular, unlike prior works like SVD-LLM and SVD-LLM v2, our theoretical framework is derived from scratch to explicitly handle redundant data (that is the main reason we make use of pseudoinverses). (By redundant, we mean data living in a nontrivial subspace, which implies singular second moments.) This case arose in our early experiments and is handled in our framework in a principled way rather than with the addition of random noise.
>
> Additionally, we provide a more general characterization of whitening matrices; our theory is agnostic to the way they are computed, which then allows us to obtain them via an eigendecomposition of a Hermitian matrix instead of an SVD, yielding a faster procedure.
>
> Finally, the objective of our $(\overline{O}, \overline{I}, \overline{P})$ framework is not only to allow immediate use in, for example, convolutional layers, but also to establish a baseline that will allow direct extensions for other layers, with the guarantee that results will hold.
>
> We believe these are significant contributions that go beyond purely engineering aspects.
>
> >**W2** Handling residual connections
>
> We agree with both points. For clarity, we will comment on the second weakness in two parts: one corresponding to the theoretical bounds and another to the practical implementation.
>
> **Practical handling of ResNet-type models.** Our framework operates at the layer level: it replaces each layer's internal computation by two sequential sub-layers, without changing input/output dimensions. As a result, it is agnostic to network structure. ResNet-style modules are just compositions of additions and convolutions. Therefore, residual connections require no specific handling.
>
> We agree that this implicit point may not be obvious to all readers and will add a short discussion to emphasize it. Thanks for pointing it out.
>
> Finally, as we mention in Appendix F, we consider every convolutional layer for decomposition. This is a user choice: one could, for instance, decide not to compress layers in residual pathways, or to force the first and/or last layer to remain uncompressed. Thanks to our rank allocator, we can freely allow every layer to be potentially compressed, and "let it decide". (In our early experimentation, we did not see benefits from manually keeping sensitive layers uncompressed.)
>
> **Model-level theoretical bounds.** We chose to state Theorem 3 for sequential networks to keep the result readable while still providing intuitive insight.
>
> However, our result is easy to extend to the residual case. We show here the main ideas for an extension to a one-step residual layer. We omit biases, since they vanish (as shown in our main results).
>
> *Case 1.* Suppose we do not use convolutions in the residual path (either because sizes match, or because we use a non-trainable approach, for instance). Then, the one-step residual layer can be expressed as
> $$
> r(X;W) = a(f(X;W)) + d(X) = a(\overline{O}(\overline{I}(X)\overline{P}(W))) + d(X),
> $$
> where $d(X)$ denotes the operation on the residual path, which is not trainable (like the identity function, or a basic downsampling operation), $f(X;W)$ is the convolutional layer, and $a$ is the activation function.
>
> We note that, for compressed weights $\hat{W}$,
> $$
> r(X;W) - r(X; \hat W) = a(f(X;W)) - a(f(X;\hat W)),
> $$
> hence the strategy to bound this case is the same as our original one (the residual paths vanish).
>
> *Case 2.* Now consider the case where the residual path also contains a convolution that will be (potentially) compressed. That is,
> $$
> r(X;W) = a_1 (f(X;W)) + a_2(g(X;W)),
> $$
> where $f(X;W)$ is the main convolution and $g(X;W)$ is in the residual pathway. One can then bound the errors as $\lVert \text{error}_r \rVert \leq \lVert \text{error}_f \rVert  + \lVert \text{error}_g \rVert $ by the triangle inequality.
>
> We still believe handling every case in the general proof would make it unnecessarily difficult to parse. However, we believe that a short remark after the formal version of the theorem is a nice addition, and we will add it to the paper.
>
> > **Q1** Can BALF be composed with post-training quantization?
>
> Yes. The resulting network has the same structure as the original, but with each layer being (potentially) replaced by two sequential layers. Those are simply convolutional or linear layers; hence, any existing quantization (or even pruning) technique can be applied without any additional adaptation.

---

> > ### Comment · Reviewer_LWbM · 2025-11-24
> > **Response to author**
> >
> > Thank you for the detailed rebuttal and clarifications, which have largely resolved my technical concerns. However, the additional discussion on innovation and generality, while helpful, is not sufficient for me to raise the score beyond 6. I also note that the updated PDF does not highlight the changes, which makes it harder to identify the revisions.

---

> ### Author Response · Authors · 2025-11-13
> **Response to Reviewer LWbM, part 2**
>
> >**Q2** BN + Conv fusion
>
> First, we note that, in the general setting (and in particular when fusion is done after compression), not fusing BN and Conv layers is algebraically equivalent to fusing them. In our main experiments, we do not fuse them, but if a practitioner were to use a compressed network for a real inference task, they could just fuse the BN and Conv layers after compressing exactly as usual, and the results would be the same. The way of fusing does not change: it is just output-channel-wise scaling and shifting.
>
> In our accuracy benchmarks, we do not correct BN statistics, so as to provide a fair evaluation against other methods. Out of curiosity, we tried it a while ago, but we did not observe consistent accuracy benefits.
>
> However, in our speed benchmarks, in order to mimic inference behavior, we actually do the equivalent of fusing Conv and BN layers, and it is done after factorization (in order to match the numerics of our accuracy experiments).
>
> ### Conclusion
>
> In summary, we propose a framework with a more general and more robust theoretical foundation, which makes our work an impactful contribution beyond purely practical and engineering considerations.
>
> Again, we thank you for bringing up those valuable points, and we hope these clarifications address your concerns.
>
> We would be happy to provide any further clarifications if needed.

---

> ### Author Response · Authors · 2025-11-24
>
> Thank you for your reply and for pointing that out, as well as for your earlier thoughtful comments and suggestions. We understand and respect the decision. If any additional suggestions or comments arise during the remainder of the discussion phase, we would be very happy to address them.

---

### Official Review · Reviewer_Zr6M · 2025-10-31

**Soundness:** 2
**Presentation:** 2
**Contribution:** 1
**Rating:** 2
**Confidence:** 5

**Summary:**

This paper presents BALF, an efficient pipeline for compressing models using low-rank matrix decomposition without fine-tuning.

**Strengths:**

+ The motivation of this paper is clear, i.e., to select the optimal ranks using SVD to factorize the matrices of the target DNN models.
+ This paper provides sufficient theoretical analysis for the proposed method.
+ The proposed method shows effectiveness on several CNN models in the experiments.

**Weaknesses:**

- Limited novelty. The idea of compressing neural networks through matrix factorization is long-established, and many prior works have investigated rank selection strategies. In addition, the use of augmented Lagrangian formulations for enforcing low-rank constraints in network compression has already been explored [R1]. The paper does not seem to introduce a fundamentally new principle or formulation.

- Lack of large-scale validation. Although the motivation centers on compressing large models, no experiments are conducted on modern large-scale architectures such as LLMs. Instead, the evaluation focuses on relatively small and dated models (e.g., ResNet, ResNeXt, ViT). Given the large body of existing work applying matrix factorization to such models, the contribution appears incremental.

- Missing comparison with alternative compression methods. The paper does not compare its method against other major compression approaches, such as pruning or quantization, making it difficult to assess the relative performance and practicality of the proposed technique.

- Insufficient baselines in experiments. The set of comparison methods in the experimental section is very limited, and several state-of-the-art approaches, e.g., [R1] mentioned in the related work section, are not included in the empirical evaluation. This weakens the paper’s claim of superiority or generality.

- Unvalidated theoretical claims. Although the paper presents several theoretical results, the lack of strong experimental evidence and comprehensive comparisons makes it hard to verify the practical value of these theoretical contributions.

[R1] Low-rank compression of neural nets: Learning the rank of each layer.

**Questions:**

Please see the Weaknesses.

---

> ### Author Response · Authors · 2025-11-13
> **Response to Reviewer Zr6M**
>
> We thank you for the detailed review and comments. We believe that several of your concerns stem from misunderstandings of our method and experiments; we clarify them point by point below.
>
> >**W1** Limited novelty
>
> **Novelty of our decomposition framework.** Our framework is not the standard matrix factorization approach. Our first contribution is a principled and general activation-aware technique based on the fact that some popular layers (such as convolutions) are expressible as matrix multiplications (as noted in Definition 1 and Examples 1 to 3) plus auxiliary operations (we call this family of layers "$(\overline{O}, \overline{I}, \overline{P})$-expressible"). We back this up with theoretical analysis; for instance, we show that, for any layer that fits our framework, our factorization scheme is optimal in the sense of layer output distortion (Theorem 1). We also provide network-level bounds on output distortion (Theorem 3).
>
>
> **Differences between [R1] and our rank allocator.** The main goal of our rank allocation technique is to be effective and precise (i.e., produce a model within a given computational constraint and with minimal accuracy loss) when compressing a model, while avoiding the overhead (for instance, additional training or expensive search) that is generally caused by other methods (e.g., [R1]). This is explained in Section 4.3.
>
> We use a Lagrange relaxation to achieve that, and it plays a fundamentally different role from the way [R1] employs Lagrange multipliers: while they use them to solve an augmented training-based optimization problem that includes rank selection, we use Lagrange relaxation to solve the combinatorial problem of selecting ranks under a compression budget, which can be seen as an instance of the multiple-choice knapsack problem. To the best of our knowledge, this has not been explored in the past.
>
> >**W2**, **W3**, **W4** Experimental section
>
> We believe these concerns are already addressed by our experimental section (Section 5), which we clarify here.
>
> **Breadth and scale.** We carry out experiments on 9 different models (including ResNet, ResNeXt, ViT, and MobileNet architectures) at different scales (including, e.g., ResNeXt-101 and ViT-B models on ImageNet). These experiments span models with ungrouped convolutions, grouped convolutions, and linear layers. Our method is architecture-agnostic, and we believe that our experiments on ViTs already showcase the effectiveness of our method on transformer architectures.
>
>
> **Comparison and baselines.** In contrast to the claim that we do not have enough baselines and that we do not compare with other compression methods, we compare against 13 methods:
> - 2 baseline factorization methods (see Figure 2 and Figure 9), which also serve to gauge the importance of the different components of our framework.
> - 11 additional methods, covering other factorization techniques, pruning, and (for ViTs) token merging (see Table 1 for these comparisons).
>
> In most model compression studies, quantization approaches are not compared directly to other compression techniques because of their fundamental differences. Put succinctly, quantization does not alter the model structure; it reduces the cost per operation and/or memory bandwidth requirements, which makes metric-based comparison challenging. As a side note, quantization is orthogonal to our method, meaning that it can be applied to factorized layers in the same way as one would apply it to the original layer.
>
> Among the methods we compare to, [R1] (which was claimed to be missing) is included where results are available (on MobileNet-V2); our method outperforms it, even without requiring any training.
>
> The general conclusion of those comparisons is that our method performs better than other compression approaches in most scenarios, and is competitive in the remaining ones.
>
> >**W5** Unvalidated theoretical claims
>
> We note that all our theoretical claims are proven rigorously (the proofs can be found in the Appendix).
>
> Theorem 1 states that the factorization framework is optimal in the sense of layer output distortion. This is also explicitly demonstrated in Figure 1. Theorem 2 simply provides a closed-form solution to the output distortion incurred when compressing a layer.
>
> On the other hand, the objective of Theorem 3 is to provide provable insight into the behavior of our method: it bounds the network-level distortion of a simple network.
>
> Moreover, the overall implications of our theoretical results are observed in our end-to-end experiments in Section 5.
>
> ### Conclusion
>
> We hope these clarifications address your concerns. In summary, our work introduces a general and theoretically grounded activation-aware factorization framework, a distinct and efficient global rank allocation procedure, and strong empirical performance across diverse architectures and baselines.
>
> We would be happy to provide any further clarifications during the discussion phase if needed.

---

> > ### Comment · Reviewer_Zr6M · 2025-11-26
> >
> > Thank you for the response. I would like to increase my rating.

---

> ### Author Response · Authors · 2025-11-26
>
> Thanks for your response to the rebuttal. It is set to private (not visible to the public), which might have been a missclick. Did we correctly address all points? If there is any concern remaining, we would be eager to help clarify it.

---

### Official Review · Reviewer_4J3Z · 2025-11-01

**Soundness:** 3
**Presentation:** 3
**Contribution:** 3
**Rating:** 8
**Confidence:** 3

**Summary:**

The paper suggests a unified and efficient fine-tuning-free compression framework for deep neural networks that combines activation-aware low-rank factorization with a budgeted rank allocation mechanism.

Traditional low-rank factorization techniques reduce model size by truncating singular values of layer weight matrices, but they often require fine-tuning or heuristic search to recover accuracy. BALF addresses this by (1) extending activation-aware decomposition to general layer types (including grouped convolutions) and (2) introducing a scalable, zero-overhead budgeted allocator that meets user-specified parameter or FLOPs constraints without retraining.

The method computes uncentered whitening matrices of layer activations to perform activation aware SVD, minimizing output distortion rather than parameter distortion. A Lagrangian relaxation formulation then determines per-layer rank allocations to meet global resource budgets efficiently.

Empirical results are also provided, outperforming existing fine-tuning-free baselines in both accuracy and runtime. On an RTX 2070, compression of ImageNet-scale models completes in minutes without fine-tuning.

**Strengths:**

Strengths

Conceptual innovation -
BALF reframes low-rank compression through an activation-centric lens, ensuring that the projection minimizes functional output distortion rather than raw parameter deviation. This distinction aligns compression with representational behavior.

General framework-The authors generalize activation-aware SVD to a more general expressible layer, encompassing dense, convolutional, and grouped convolutional layers in a unified algebraic formulation.

Mathematical rigor: Theorems 1 and 2 show equivalence between activation aware and direct output truncation schemes, with a closed-form expression for activation distortion in terms of singular values.

I especially likes the rank allocator that transforms a combinatorial multiple choice knapsack problem into a linear-time Lagrangian relaxation, which enables control over global compression budgets.

Empirical performance: The improvement is consistent compared to standard SVD baselines, maintaining high accuracy at large compression ratios.

Reproducibility and presentation:
Implementation details, pseudocode, and open-source repository are provided, which enable accessibility for less theory researchers.

**Weaknesses:**

- The proof assumes ungrouped layers and bounded Lipschitz constants without empirical validation.
- The theoretical bounds may be too loose for larger networks.
- The method is not so strong on already-optimized architectures (e.g., MobileNet-V2). The adjustment might be adapted for each architecture.
The description of FLOP estimation and calibration sampling could be expanded for full reproducibility.

**Questions:**

The theoretical bounds depend on Lipschitz constants. Can't they be too high in practice?

---

> ### Author Response · Authors · 2025-11-13
> **Response to Reviewer 4J3Z**
>
> We are really grateful for the thorough review, the positive evaluation, and the valuable concerns you raised. We will do our best to address the different points.
>
> > **W1** The proof assumes ungrouped layers and bounded Lipschitz constants without empirical validation.
>
> **Ungrouped layer assumption.**  In a similar vein to our response to Reviewer LWbM on residual connections, we believe that this is more a matter of balance between the complexity of the result and the insight it provides, and we chose to keep the ungrouped version. It is possible to extend it to the grouped case by bounding the "batched" versions of the expressions instead; however, the result gets convoluted and is harder to grasp, without providing additional insight.
>
> **Lipschitz constants.**  We emphasize that the Lipschitz constants used explicitly in the text are fixed (i.e., not unbounded) and generally not the cause of looseness. The first Lipschitz constants we explicitly declare are those of activation functions, which are generally 1 for common activation functions (e.g., ReLU or GELU). The remaining ones are the Lipschitz constants for the $\overline{I}$ functions; for example, for linear layers, the constant is also 1.
>
> > **W2** The theoretical bounds may be too loose for larger networks.
>
> This is indeed a valid point, and we agree with you. As we acknowledge in the text, the bound is loose in deep networks. As noted in Appendix E, some works aim to make networks "unitary", which could tighten our bounds (we mention this as a curiosity, as typical networks do not follow this principle).
>
> However, we note that the main causes of this issue are not the Lipschitz constants explicit in the proof but the spectral norms (i.e., the "implicit" Lipschitz constants) of the matrices involved.
>
> Moreover, we do not intend the bound to be a precise predictor, but rather an informative result that provides insight into how our framework behaves. Importantly, we never rely on this bound for rank selection in practice.
>
> We believe that the last question (**Q1**) is subsumed by **W1** and **W2**.
>
> > **W3** Already-optimized networks and experimental details
>
> **Compact networks.** We agree with you. We note, however, that this is a prevalent issue with compression methods in already-compact networks. Since they already make good use of their parameters and/or compute, removing components becomes challenging. We note that, on MobileNet-V2, BALF is only outperformed (in terms of parameter compression) by ALDS (see Table 1). We would appreciate it if you could clarify what you meant by "The adjustment might be adapted for each architecture".
>
> **Experimental details.** These details are included in the manuscript. The exact FLOP counts for each layer (pre- and post-factorization) are in Appendix A, and the way we sample calibration data is discussed in Appendix F (for all our experiments, we use uniform sampling). Moreover, if you are curious, the manuscript also includes a section (Appendix G) that discusses the effect of different calibration sizes.
>
> ### Conclusion
>
> Again, we thank you for bringing up those valuable points, and we hope our responses correctly addressed your questions and concerns.
>
> We would be happy to provide any further clarifications if needed.

---

### Author Response · Authors · 2025-11-21
**We have uploaded a revised manuscript**

We are pleased to communicate that we have uploaded a revised version of our manuscript incorporating solutions to the reviewers' concerns. Below, we specify the changes made. In general, these changes reflect our existing responses to each reviewer's comments and questions.

**On Reviewer 4J3Z's comments.** In Appendix D, we have added clarifications regarding the source of the bound looseness and provided additional discussion of the grouped case. We have also included a short paragraph in Appendix E.3 detailing how we count FLOPs and parameters.

**On Reviewer LWbM's comments.** In the contributions summary in Section 1, we now explicitly mention that we rework the theory to handle possibly redundant data. In Section 4.4, we have added a short discussion emphasizing the layer-level nature of our framework and its easy composition with other compression methods. In Appendix E.3, we explicitly address the question of BN + Conv fusion. In Appendix D, we have included a brief discussion of how to extend the bounds to residual layers, consistent with our response.

**On Reviewer Zr6M's comments.** We believe these points were already addressed in the previous version of the paper (as mentioned in our rebuttal response), but we are happy to follow up if the reviewer feels otherwise.

**Additional updates.** We have updated our notion of whitening matrices (as defined in Section 3) to include an additional condition (which holds in practice and does not change any theoretical result) that is required for a property used in our optimality proof to hold. We emphasize that the implications of said change are minimal; our practical setting still matches the new notion of whitening matrices and the theoretical results remain unchanged. We also corrected the tables and figures regarding the results on the ViT architecture, which previously contained an error affecting FLOP counts (we have recomputed the FLOP counts for these models; the accuracy and parameter counts remain unchanged). This error is negligible (less than a 2 percentage-point change in FLOP ratios) and is not even perceptible in the figures.

Finally, we made several secondary clarification edits where needed, and numerous style changes that improve presentation but do not affect the paper's content. We want to note that, due to the style changes, the appendix section letters have shifted upwards. (E.g., the old Appendix B is now Appendix A, and so on.)

**Remarks.** Overall, we addressed all reviewers' concerns with targeted clarifications, added discussions, and improved the presentation of our paper. No core claims or results were changed.

Again, we are very grateful for the time and effort of the reviewers, which has led to actual improvements in the clarity of our paper. We are still eager to answer any subsequent question or concern that might arise.

---

### Author Response · Authors · 2025-12-03
**Clarification**

Using an updated version of our code in another project, but with the same core algorithm, we have recently observed some performance drops in perplexity relative to SVD-LLM on LLMs. We currently attribute this to the difference discussed in Section 4.4: the impact of not updating whitening matrices after compressing layers appears to be larger in LLMs. In a preliminary result (to be taken with a grain of salt and which might not be completely fair with respect to SVD-LLM), we notice an increase from 8.5 (reported by SVD-LLM) to 11.3 in perplexity on Llama 2-7B at 20% compression in terms of parameter count (in the fine-tuning-free setting).

This drop comes with the benefit of substantially lower compression time when generating models at multiple compression ratios (as discussed in Appendix G and briefly in Section 5, our method allows caching and reusing factors and whitening matrices, while SVD-LLM's updating strategy does not).

We note that every result presented in the paper remains correct. We will add this information explicitly in the limitations section.

---

### Meta-Review · Area_Chair_E4B5 · 2025-12-27

**Summary:**

This paper proposes a network compression approach that combines activation-aware low-rank projection on batched matrices with a layer-wise rank allocation strategy under a global budget constraint.

A central concern raised by multiple reviewers is the level of methodological novelty. The main components of the proposed approach—including activation-aware factorization, layer-wise rank selection, and the use of Lagrangian-based budgeted optimization—have all appeared in prior work. As such, the contribution is better characterized as a systems- or pipeline-level integration rather than a clear methodological breakthrough.

Given this positioning, a stronger empirical case is required to justify the value of the proposed method. However, the experimental results do not provide convincing evidence that the approach consistently outperforms closely related baselines, in particular ALDS, which also employs a budgeted layer-wise rank allocator. The reported tables do not include sufficiently controlled or matched comparisons (e.g., at comparable compression ratios or accuracy levels) to establish a clear advantage.

If the intended contribution is improved efficiency or scalability rather than superior accuracy–compression trade-offs, a more explicit discussion and evaluation along these dimensions (e.g., runtime, stability, or implementation simplicity relative to ALDS) would be necessary to clarify the benefits of the approach.

Overall, due to the limited methodological novelty and the lack of compelling empirical evidence demonstrating advantages over similar existing methods, I do not recommend acceptance at this time.

**Reviewer Concerns:**

Reviewers’ questions regarding technical details were largely addressed in the rebuttal. However, concerns about methodological novelty and the strength of the proposed approach relative to existing baselines remain outstanding.

**Reviewer Scores:**

Reviewer 4J3Z is unlikely to increase the score, as the initial rating (8) is already very high.

Reviewer Zr6M gave an initial score of 2. In the rebuttal discussion, the reviewer indicated that the rating would be increased, but did not explicitly state that the core concerns had been resolved. Given that the primary concern relates to novelty, it is unlikely that the score would increase substantially.

Reviewer LWbM gave an initial score of 6 and explicitly indicated that the score would not be increased further following the rebuttal.

---

### Decision · Program_Chairs · 2026-01-26

Reject